# CausalRAG2: Hierarchical Causal Knowledge Graph Design for RAG

**Nengbo Wang** [1 2]   **Tuo Liang** [1]   **Vikash Singh** [1]   **Chaoda Song** [1]   **Van Yang** [1]
**Yu Yin** [1]   **Jing Ma** [1]   **Jagdip Singh** [2]   **Vipin Chaudhary** [1]

## Abstract

Retrieval augmented generation (RAG) has enhanced large language models by enabling access to external knowledge, with graph-based RAG emerging as a powerful paradigm for structured retrieval and reasoning. However, existing graph-based methods often over-rely on entity-centric node matching and lack explicit causal modeling, leading to unfaithful or spurious answers. Prior attempts to incorporate causality are typically limited to local or single-document contexts and also suffer from information isolation that arises from modular graph structures, which hinders scalability and cross-module causal reasoning. To address these challenges, we propose CausalRAG2, a framework that rethinks knowledge organization for graph-based RAG through causal gating across hierarchical modules. CausalRAG2 explicitly models causal relationships to suppress spurious correlations while enabling scalable reasoning over large-scale knowledge graphs. We also introduce HolisQA, a benchmark for holistic comprehension beyond entity-centric matching. Extensive experiments demonstrate that CausalRAG2 consistently outperforms competitive graph-based RAG baselines across multiple datasets and evaluation metrics. Our work establishes a principled foundation for structured, scalable, and causally grounded RAG systems. Our code and HolisQA benchmark are available at https://github.com/Pwnb/CausalRAG2.

## 1. Introduction

While Retrieval-Augmented Generation (RAG) effectively extends Large Language Models (LLMs) with external knowledge (Lewis et al., 2020), traditional pipelines predominantly rely on text chunking and semantic embedding search. This paradigm implicitly frames knowledge access as a flat similarity matching problem, overlooking the structured and interdependent nature of real-world concepts. Consequently, as knowledge bases scale in complexity, these methods struggle to maintain retrieval efficiency and reasoning fidelity.

Graph-based RAG has emerged as a promising solution to address these gaps, led by frameworks like GraphRAG (Edge et al., 2025) and extended through agentic search (Ravuru et al., 2024), GNN-guided refinement (Liu et al., 2025a), and hypergraph representations (Luo et al., 2025a). However, three unintended limitations still persist. First, current research prioritizes retrieval policies while overlooking knowledge graph organization. As graphs scale, intrinsic modularity (Fortunato & Barthélemy, 2007) often restricts exploration within dense modules, triggering **information isolation.** Common grouping strategies ranging from communities (Edge et al., 2025), passage nodes (Gutiérrez et al., 2025), node-edge sets (Guo et al., 2025) to semantic grouping (Zhang et al., 2026) often inadvertently reinforce these boundaries, severely limiting global recall. Second, most formulations rely on semantic proximity and superficial traversal on graphs without **causal awareness**, leading to a **locality issue** where spurious nodes and irrelevant noise degrade precision (see Figure 1). Despite the inherent causal discovery potential of LLMs, this capability remains largely untapped for filtering noise within RAG pipelines. Finally, these systemic flaws are often masked by popular QA dataset evaluations, which reward entity-level "hits" over holistic comprehension. Consequently, there is a pressing need for a retrieval framework that **reconciles global knowledge accessibility with local reasoning precision** to support robust, causally grounded generation.

To address these challenges, we propose CausalRAG2, a framework that rethinks knowledge graph organization through **hierarchical causal gate structures**. CausalRAG2 formulates the knowledge graph as a multi-layered repre-

---

[1]Department of Computer and Data Sciences, Case Western Reserve University, Cleveland, OH, USA [2]Design and Innovation Department, Case Western Reserve University, Cleveland, OH, USA. Correspondence to: Nengbo Wang <nxw189@case.edu>, Vipin Chaudhary <vxc204@case.edu>.

*Proceedings of the 43rd International Conference on Machine Learning*, Seoul, South Korea. PMLR 306, 2026. Copyright 2026 by the author(s).

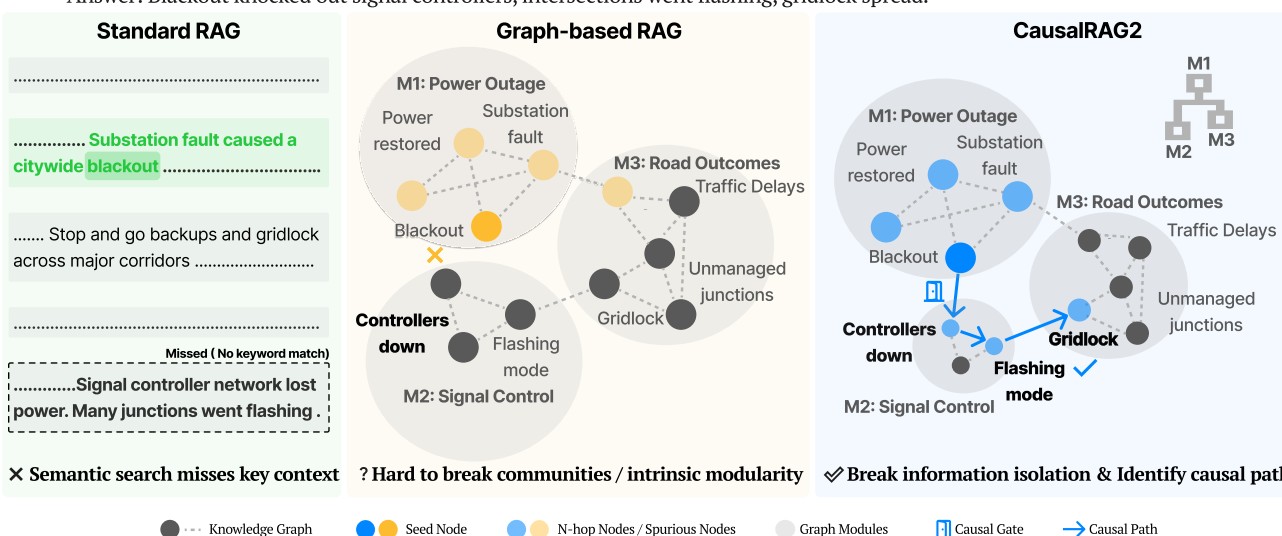

*Figure 1.* Comparison of three retrieval paradigms, Standard RAG, Graph-based RAG, and CausalRAG2, on a citywide blackout query. Standard RAG misses key evidence under semantic retrieval. Graph-based RAG can be trapped by intrinsic modularity or grouping structure. CausalRAG2 leverages hierarchical causal gates to bridge modular boundaries, effectively breaking information isolation and explicitly identifying the underlying causal path.

sentation where fine-grained facts are organized into higher-level schemas, enabling multi-granular reasoning. This hierarchical architecture, integrated with causal gates, establishes logical bridges across modules, thereby naturally breaking information isolation and enhancing global recall. During retrieval, CausalRAG2 moves beyond point-wise semantic matching to explicit reasoning over causal graphs. By actively distinguishing genuine causal dependencies from spurious associations, CausalRAG2 mitigates the locality issue and filters retrieval noise to ensure precise, grounded, and interpretable generation.

To validate the effectiveness of CausalRAG2, we conduct extensive evaluations across datasets in multiple domains, comparing it against a diverse suite of competitive RAG baselines. To address the previously identified limitations of existing QA datasets, we introduce a large-scale cross-domain dataset HolisQA focused on **holistic comprehension**, designed to evaluate reasoning capabilities in complex, real-world scenarios. Our results consistently demonstrate that causal gating and causal reasoning effectively reconcile the trade-off between recall and precision, significantly enhancing retrieval quality and answer reliability.

## 2. Related Work

### 2.1. RAG

Retrieval augmented generation grounds LLMs in external knowledge, but chunk level semantic search can be brittle and inefficient for large, heterogeneous, or structured corpora (Lewis et al., 2020). Graph-based RAG has therefore emerged to introduce structure for more informed retrieval.

**Graph-based RAG.** GraphRAG constructs a graph structured index of external knowledge and performs query time retrieval over the graph, improving question focused access to large-scale corpora (Edge et al., 2025). Building on this paradigm, later work studies richer selection mechanisms over structured graph. Agent driven retrieval explores the search space iteratively (Ravuru et al., 2024). Critic-guided or winnowing style methods prune weak contexts after retrieval (Dong et al., 2025a; Wang et al., 2025b). Others learn relevance scores for nodes, subgraphs, or reasoning paths, often with graph neural networks (Liu et al., 2025a). Representation extensions include hypergraphs for higher order relations (Luo et al., 2025a) and graph foundation models for retrieval and reranking (Wang et al., 2026b).

**Knowledge Graph Organization.** Despite these advances, limitations related to graph organization remain underexamined. Most work emphasizes retrieval policies, while the organization of the underlying knowledge graph is largely overlooked, which strongly influences downstream retrieval behavior. As graphs scale, intrinsic modularity can emerge (Fortunato & Barthélemy, 2007; Newman, 2018), making retrieval prone to staying within dense modules rather than crossing them, largely limiting the retrieved information. Moreover, many works assume grouping knowledge for efficiency at scale, such as communities (Edge et al., 2025), phrases and passages (Gutiérrez et al., 2025), node-

| Method | Knowledge Graph Organization | Retrieval and Generation Process |
|---|---|---|
| Standard RAG (Lewis et al., 2020) | Flat text chunks, unstructured. $\mathcal{G}_{\text{idx}} = \{d_i\}_{i=1}^N$ | Semantic vector search over chunks. $S = \text{TopK}(\text{sim}(q, d_i)); \quad y = \mathsf{G}(q, S)$ |
| GraphRAG (Edge et al., 2025) | Partitioned communities with summaries. $\mathcal{G}_{\text{idx}} = \{\text{Sum}(c) \mid c \in \mathcal{C}\}$ | **Map-Reduce** over community summaries. $A_{\text{part}} = \{\mathsf{G}(q, \text{Sum}(m))\}; \quad y = \mathsf{G}(A_{\text{part}})$ |
| LightRAG (Guo et al., 2025) | Dual-level indexing (Entities + Relations). $\mathcal{G}_{\text{idx}} = (V_{\text{ent}} \cup V_{\text{rel}}, E)$ | **Keyword-based** vector retrieval + neighbor. $K_q = \text{Key}(q); \quad S = \text{Vec}(K_q, \mathcal{G}_{\text{idx}}) \cup \mathcal{N}_1$ |
| HippoRAG 2 (Gutiérrez et al., 2025) | Dense-sparse integration (Phrase + Passage). $\mathcal{G}_{\text{idx}} = (V_{\text{phrase}} \cup V_{\text{doc}}, E)$ | **PPR** diffusion from LLM-filtered seeds. $U_{\text{seed}} = \text{Filter}(q, V); \quad S = \text{PPR}(U_{\text{seed}}, \mathcal{G}_{\text{idx}})$ |
| LeanRAG (Zhang et al., 2026) | Hierarchical semantic clusters (GMM). $\mathcal{G}_{\text{idx}} = \text{Tree}(\text{Semantic Aggregation})$ | **Bottom-up** traversal to LCA (Ancestor). $U = \text{TopK}(q, V); \quad S = \text{LCA}(U, \mathcal{G}_{\text{idx}})$ |
| CausalRAG (Wang et al., 2025a) | Flat graph structure. $\mathcal{G}_{\text{idx}} = (V, E)$ | Top-K retrieval + Implicit causal reasoning. $S = \text{Expand}(\text{TopK}(q, V)); \quad y = \mathsf{G}(q, S)$ |
| **CausalRAG2 (Ours)** | **Hierarchical Causal Gates** across modules. $\mathcal{G}_{\text{idx}} = \mathcal{H} = \{H_0, \ldots, H_L\}$ | **Causal Gating + Causal Path Filtering**. $S = \underbrace{\text{Traverse}(q, \mathcal{H})}_{\text{Break Isolation}} \cap \underbrace{\text{Filter}_{\text{causal}}(S)}_{\text{Reduce Noise}}$ |

*Table 1.* Comparison of RAG frameworks based on knowledge organization and retrieval mechanisms. **Notation:** $\mathcal{M}$ modules, Sum($\cdot$) summary, PPR Personalized PageRank, $\mathcal{H}$ hierarchy, $\mathcal{N}_1$ 1-hop neighborhood.

edge sets (Guo et al., 2025), or semantic aggregation (Zhang et al., 2026) (see Table 1), which can amplify modular confinement and yield **information isolation**. This **global issue** primarily manifests as **reduced recall**. Some hierarchical approaches like LeanRAG attempt to bridge these gaps via semantic aggregation, but they remain constrained by semantic clustering and rely on tree-structured traversals (Zhang et al., 2026), often failing to capture logical dependencies that span across semantically distinct clusters.

**Retrieval Issue.** A second limitation concerns how retrieval is formulated. Much work operates as a multi-hop search over nodes or subgraphs (Gutiérrez et al., 2025; Liu et al., 2025b), prioritizing semantic proximity to the query without explicit awareness of the reasoning in this searching process. This design can pull in topically similar yet causally irrelevant evidence, producing conflated retrieval results. Even when the correct fact node is present, the generator may respond with generic or superficial content, and the extra noise can increase the risk of hallucination. We view this as a **locality issue that lowers precision**.

**QA Evaluation Issue.** These tendencies can be reinforced by common QA evaluation practice. First, many QA datasets emphasize short answers such as names, nationalities, or years (Kwiatkowski et al., 2019; Rajpurkar et al., 2016), so **hitting the correct entity** in the graph may be sufficient even without reasoning. Second, QA datasets often comprise thousands of independent question-answer-context triples. However, many approaches still rely on linear context concatenation to construct a graph, and then evaluate performance on isolated questions. This setup **largely reduces the incentive for holistic comprehension** of the underlying material, even though such end-to-end understanding is closer to real-world use cases. Third, some

datasets are stale enough that answers may be partially memorized by pretrained LLM models, **confounding retrieval quality with parametric knowledge**. Therefore, these QA dataset issues are critical for evaluating RAG, yet relatively few works explicitly address them by adopting open-ended questions and fresher materials in controlled experiments.

## 2.2. Causality

**LLM for Identifying Causality.** LLMs have demonstrated exceptional potential in causal discovery. By leveraging vast domain knowledge, LLMs significantly improve inference accuracy compared to traditional methods (Ma, 2025). Frameworks like CARE further prove that fine-tuned LLMs can outperform state-of-the-art algorithms (Dong et al., 2025b). Crucially, even in complex texts, LLMs maintain a direction reversal rate under 1.1% (Saklad et al., 2026), ensuring highly reliable results.

**Causality and RAG.** While LLMs increasingly demonstrate reliable causal reasoning capabilities, explicitly integrating causal structures into RAG remains largely underexplored. Current research predominantly focuses on internal attribution graphs for model interpretability (Walker & Ewetz, 2025; Dai et al., 2026), rather than external knowledge retrieval. Recent advances like CGMT (Luo et al., 2025b) and LACR (Zhang et al., 2024) have begun to bridge this gap, utilizing causal graphs for medical reasoning path alignment or constraint-based structure induction. However, these works inherently differ in scope from our objective, as they prioritize rigorous causal discovery or recovery tasks in a specific domain, which limits their scalability to the noisy, open-domain environments that we address. Existing causal-enhanced RAG frameworks either utilize causal feedback implicitly in embedding (Khatibi et al., 2025) or,

like CausalRAG (Wang et al., 2025a), are restricted to small-scale settings with implicit causal reasoning. Consequently, a significant gap persists in leveraging causal graphs to guide knowledge graph organization and retrieval across large-scale, heterogeneous knowledge bases. Note that in this work, we use the term *causal* to denote explicit logical dependencies and event sequences described in the text, rather than statistical causal discovery from observational data.

## 3. Problem Formulation

We aim to retrieve an optimal subgraph $S^\star \subseteq \mathcal{G}$ for a query $q$ to generate an answer $y$. Graph-based RAG ($S = \mathcal{R}(q, \mathcal{G})$) usually faces two structural bottlenecks.

**1. Global Information Isolation (Recall Gap).** Intrinsic modularity often traps retrieval in local seeds, missing relevant evidence $v^*$ located in topologically distant modules (i.e., $S \cap \{v^*\} = \emptyset$ as no path exists within $h$ hops). **CausalRAG2** introduces *causal gates* across $\mathcal{H}$ to bypass modular boundaries and bridge this gap. The efficacy of causal gates is empirically verified in Appendix E and further analyzed in the ablation study (see Section 5.3).

**2. Local Spurious Noise (Precision Gap).** Semantic similarity $\text{sim}(q, v)$ often retrieves topically related but causally irrelevant nodes $\mathcal{V}_{sp}$, diluting precision (where $|S \cap \mathcal{V}_{sp}| \gg |S \cap \mathcal{V}_{causal}|$). We address this by leveraging LLMs to identify explicit *causal paths*, filtering $\mathcal{V}_{sp}$ to ensure groundedness. While as discussed LLMs have demonstrated causal identification capabilities surpassing human experts (Ma, 2025; Dong et al., 2025b) and proven effectiveness in RAG (Wang et al., 2025a), we further corroborate the validity of identified causal paths through expert knowledge across different domains (see Section 5.1). Consequently, CausalRAG2 redefines retrieval as finding a mapping $\Phi : \mathcal{G} \to \mathcal{H}$ and a causal filter $\mathcal{F}_c$ to simultaneously minimize isolation and spurious noise.

## 4. Method

**Overview.** As illustrated in Figure 2, CausalRAG2 operates in two distinct phases to address the aforementioned structural bottlenecks. In the **offline phase**, we construct a hierarchical knowledge structure $\mathcal{H}$ partitioned into modules, which are then interconnected via **causal gates** $\mathcal{G}_c$ to enable logical traversals. In the **online phase**, CausalRAG2 performs a **gated expansion** to break modular isolation, followed by a **causal filtering** step to eliminate spurious noise. The overall procedure is formalized in Algorithm 1 (see function definitions in Appendix B), and we detail each component in the subsequent sections.

---

**Algorithm 1** CausalRAG2 Algorithm Pipeline

---

**Require:** Corpus $\mathcal{D}$, query $q$, hierarchy levels $L$, seed budget $\{K_\ell\}_{\ell=0}^L$, hop $h$
**Ensure:** Answer $y$, Support Subgraph $S^\star$
1: *// Phase 1: Offline Hierarchical Organization*
2: $G_0 = (V_0, E_0) \leftarrow \text{BUILDBASEGRAPH}(\mathcal{D})$
3: $\mathcal{H} = \{H_0, \ldots, H_L\} \leftarrow \text{LEIDENPARTITION}(G_0, L)$ {Organize into modules $\mathcal{M}$}
4: $\mathcal{G}_c \leftarrow \emptyset$
5: **for all** pair $\{m_i, m_j\} \in \text{MODULEPAIRS}(\mathcal{M})$ **do**
6:    **if** LLM-ESTCAUSAL$(m_i, m_j) = 1$ **then**
7:       $\mathcal{G}_c \leftarrow \mathcal{G}_c \cup \{\{m_i, m_j\}\}$ {Establish undirected binary gate}
8:    **end if**
9: **end for**
10: *// Phase 2: Online Retrieval & Reasoning*
11: $U \leftarrow \bigcup_{\ell=0}^L \text{TopK}(\text{sim}(q, u), K_\ell, H_\ell)$ {Multi-level semantic seeding}
12: $S_{raw} \leftarrow \text{GATEDTRAVERSAL}(U, \mathcal{H}, \mathcal{G}_c, h)$ {Break isolation via gates}
13: $S^\star \leftarrow \text{CAUSALFILTER}(q, S_{raw})$ {Remove spurious nodes $\mathcal{V}_{sp}$}
14: $y \leftarrow \text{LLM-GENERATE}(q, S^\star)$

---

### 4.1. Hierarchical Graph with Causal Gating

To address the *global information isolation* challenge (Section 3), we construct a multi-scale knowledge structure that balances global retrieval recall with local precision.

**Hierarchical Module Construction.** We first extract a base entity graph $G_0 = (V_0, E_0)$ from the corpus $\mathcal{D}$ using an information extraction pipeline (see details in Appendix B.1), followed by entity canonicalization to resolve aliasing. To establish the hierarchical backbone $\mathcal{H} = \{H_0, \ldots, H_L\}$, we iteratively partition the graph into **modules** using the Leiden algorithm (Traag et al., 2019), which optimizes modularity to identify tightly-coupled semantic regions. Formally, at each level $\ell$, nodes are partitioned into modules $\mathcal{M}_\ell = \{m_1^{(\ell)}, \ldots, m_k^{(\ell)}\}$. For each module, we generate a natural language summary to serve as a coarse-grained semantic anchor.

**Offline Causal Gating.** While hierarchical modularity improves efficiency, it risks trapping retrieval within local boundaries. We introduce **Causal Gates** to explicitly model cross-module affordances. Instead of fully connecting the graph, we construct a sparse gate set $\mathcal{G}_c$. Specifically, we identify candidate module pairs $(m_i, m_j)$ that are topologically distant but potentially logically related. An LLM then evaluates the plausibility of a causal connection between their summaries. We formally define the gate set via an

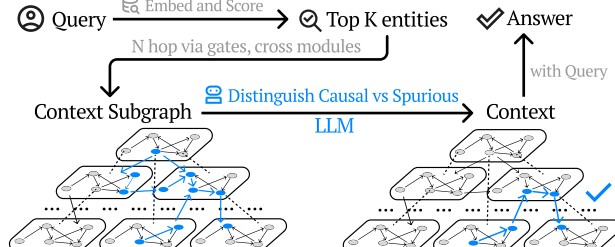

**Graph Construction (Offline)**

Raw Texts → Knowledge Graph → Vector Store

Hierarchical Graph — Identify Causality LLM → Graph with Causal Gates

**Retrieve and Answer (Online)**

Query → Top K entities → Answer

Context Subgraph — Distinguish Causal vs Spurious LLM → Context

*Figure 2.* **Overview of the CausalRAG2 pipeline.** In the offline stage, raw texts are embedded to build a knowledge graph and a vector store, then partitioning forms a hierarchical graph and an LLM identifies causal relations to construct a graph with causal gates. In the online stage, the query is embedded and scored to retrieve top-K entities, then N-hop traversal uses causal gates to cross modules and assemble a context subgraph; an LLM further distinguishes causal versus spurious relations to produce the final context and answer.

indicator function $\mathbb{I}(\cdot)$:

$$\mathcal{G}_c = \{\{m_i, m_j\} \mid \mathbb{I}_{\text{causal}}(m_i, m_j) = 1,\ i \neq j\}, \quad (1)$$

where $\mathbb{I}_{\text{causal}}$ denotes the LLM's assessment (see Appendix B.1 for construction prompts and the Top-Down Hierarchical Pruning strategy we employed to mitigate the $O(N^2)$ evaluation complexity). These undirected and binary gates act as *shortcuts* in the retrieval space, permitting the traversal to jump across disjoint modules only when logically warranted, thereby breaking information isolation without causing semantic drift (see Appendix C for visualizations of hierarchical modules and causal gates). To verify that this mechanism captures genuine causal links rather than acting as generic cross-module shortcuts, we conduct experiments comparing against random and semantic gate variants and report an expert audit of constructed causal gates in Appendix E.1.

Theoretically, causal gates address a formally definable structural bottleneck: in a modular graph $\mathcal{H}$, if seed nodes lie in module $m_i$ while gold evidence resides in a distant module $m_j$, local $h$-hop traversal is provably unable to reach $m_j$ (the information isolation problem in Section 3). Causal gates add selective gates that could reduce inter-module distance to $O(1)$, breaking isolation along only causally warranted paths.

### 4.2. Retrieve Subgraph via Causally Gated Expansion

Given the hierarchical structure $\mathcal{H}$ and causal gates $\mathcal{G}_c$, CausalRAG2 retrieves a support subgraph $S$ by coupling multi-granular anchoring with a topology-aware expansion. This process is designed to maximize recall (breaking isolation) while suppressing drift (controlled locality).

**Multi-Granular Hybrid Seeding.** Graph-based RAG often struggles to effectively differentiate between local details and global contexts within multi-level structures (Zhang et al., 2026; Edge et al., 2025). We overcome this by identifying a seed set $U$ across multiple levels of the hierarchy.

We employ a hybrid scoring function $s(q, v)$ that interpolates between semantic embedding similarity and lexical overlap (details in Appendix B.2). This function is applied simultaneously to fine-grained entities in $H_0$ and coarse-grained module summaries in $H_{\ell>0}$. Crucially, to prevent the *semantic redundancy* problem where seeds cluster in a single redundant neighborhood, we apply a diversity-aware selection strategy (MMR) to ensure the initial seeds $U$ cover distinct semantic facets of the query. This yields a set of anchors that serve as the starting nodes for expansion.

**Gated Priority Expansion.** Starting from the seed set $U$, we model retrieval as a priority-based traversal over a unified edge space $\mathcal{E}_{\text{uni}}$. This space integrates three distinct types of connectivity: (1) **Structural Edges** ($E_{\text{struc}}$) for local context, (2) **Hierarchical Edges** ($E_{\text{hier}}$) for vertical drill-down, and (3) **Causal Gates** ($\mathcal{G}_c$) for cross-module reasoning.

$$\mathcal{E}_{\text{uni}} = E_{\text{struc}} \cup E_{\text{hier}} \cup \mathcal{G}_c. \quad (2)$$

The expansion follows a Best-First Search guided by a query-conditioned gain function. For a frontier node $v$ reached from a predecessor $u$ at hop $t$, the gain is defined as:

$$\text{Gain}(v) = s(q, v) \cdot \gamma^t \cdot w(\text{type}(u, v)), \quad (3)$$

where $\gamma \in (0, 1)$ is a standard decay factor to penalize long-distance traversal. The weight function $w(\cdot)$ adjusts traversal priorities: we simply assign higher importance to causal gates and hierarchical links to encourage logic-driven jumps over random structural walks. By traversing $\mathcal{E}_{\text{uni}}$, CausalRAG2 prioritizes paths that drill down (via $E_{\text{hier}}$), explore locally (via $E_{\text{struc}}$), or leap to a causally related domain (via $\mathcal{G}_c$), effectively **breaking modular isolation**. The expansion terminates when the gain drops below a threshold or the token budget is exhausted.

### 4.3. Causal Path Identification and Grounding

The raw subgraph $S_{raw}$ retrieved via gated expansion **optimizes for recall but inevitably includes spurious**

| Datasets | Nodes | Edges | Modules | Size (Char) | Domain |
|---|---|---|---|---|---|
| MS MARCO (Bajaj et al., 2018) | 3,403 | 3,107 | 446 | 1,557,990 | Web |
| NQ (Kwiatkowski et al., 2019) | 5,579 | 4,349 | 505 | 767,509 | Wikipedia |
| 2WikiMultiHopQA (Ho et al., 2020) | 10,995 | 8,489 | 1,088 | 1,756,619 | Wikipedia |
| QASC (Khot et al., 2020) | 77 | 39 | 4 | 58,455 | Science |
| HotpotQA (Yang et al., 2018) | 20,354 | 15,789 | 2,359 | 2,855,481 | Wikipedia |
| HolisQA-Biology | 1,714 | 1,722 | 165 | 1,707,489 | Biology |
| HolisQA-Business | 2,169 | 2,392 | 292 | 1,671,718 | Business |
| HolisQA-CompSci | 1,670 | 1,667 | 158 | 1,657,390 | Computer Science |
| HolisQA-Medicine | 1,930 | 2,124 | 226 | 1,706,211 | Medicine |
| HolisQA-Psychology | 2,019 | 1,990 | 211 | 1,751,389 | Psychology |

*Table 2.* Statistics of the datasets used in evaluation.

**associations** (e.g., high-degree hubs or coincidental co-occurrences). To address the *local spurious noise* challenge (Section 3), CausalRAG2 employs a causal path refinement stage to directly distill $S_{raw}$ into a causally grounded graph $S^\star$. See Appendix D.1 for a full example of the Causal-RAG2 pipeline.

**Causal Path Refinement.** We formulate the path refinement task as a structural pruning process. We first linearize the subgraph $S_{raw}$ into a token-efficient table where each node and edge is mapped to a unique short identifier (see Appendix B.3). The LLM is then prompted to analyze the topology and output the subset of identifiers that constitute valid causal paths connecting the query to the potential answer. Leveraging the robust causal identification capabilities of LLMs (Saklad et al., 2026), this operation effectively functions as a reranker, distilling the noisy subgraph into an explicit causal structure:

$$S^\star = \text{CAUSALFILTER}(q, S_{raw}).  \quad (4)$$

The returned subgraph $S^\star$ contains only model-validated nodes and edges, effectively filtering irrelevant context.

**Spurious-Aware Grounding.** To further improve precision, we employ a **spurious-aware prompting strategy** (Appendix A.1) that instructs the LLM to explicitly distinguish causal supports from spurious correlations during reasoning. This explicit contrast helps the model resist *hallucinated connections* induced by semantic similarity, yielding a cleaner $S^\star$. Unlike a relevance reranker that produces a ranked list, the causal filter outputs a partitioned set distinguishing causal supports from spurious associations over the candidate subgraph. The answer $y$ is then generated by conditioning the LLM on the text content of $S^\star$ (Appendix A.2), ensuring grounding in verified evidence.

# 5. Experiments

**Overview.** We conducted extensive experiments on diverse datasets across various domains to comprehensively evaluate and compare the performance of CausalRAG2

against competitive baselines. Our analysis is guided by the following six research questions:

**RQ1 (Overall Performance).** How does CausalRAG2 compare against state-of-the-art graph-based baselines across diverse knowledge domains?

**RQ2 (QA vs. Holistic Comprehension).** Do popular QA datasets implicitly favor the entity-centric retrieval paradigm, inflating graph-based RAG that finds the right node without assembling a support chain?

**RQ3 (Trade-off Reconciliation).** Can CausalRAG2 simultaneously improve Context Recall (Globality) and Answer Relevancy (Precision)?

**RQ4 (Ablation Study).** What are the individual contributions of different components?

**RQ5 (Scalability Robustness).** How does performance scale and remain robust under varying context lengths?

**RQ6 (Cost Analysis).** What is the computational cost in offline construction and online queries?

## 5.1. Experimental Setup

**Datasets.** We evaluate CausalRAG2 on five established datasets: **MS MARCO** (Bajaj et al., 2018) and **Natural Questions** (Kwiatkowski et al., 2019) emphasize large-scale open-domain retrieval; **HotpotQA** (Yang et al., 2018) and **2WikiMultiHop** (Ho et al., 2020) require evidence aggregation; and **QASC** (Khot et al., 2020) targets compositional scientific reasoning. However, these datasets often suffer from entity-centric biases and potential data leakage. To test the holistic understanding capability of models, we introduce **HolisQA**, a dataset derived from high-quality academic papers (Priem et al., 2022). Spanning over diverse domains (including Biology, Computer Science, Medicine, etc.), HolisQA features dense logical structures that naturally demand holistic comprehension (see more details in Appendix F.2). All dataset statistics are summarized in Table 2. While LLMs have demonstrated strong capabilities in identifying causality (Ma, 2025; Dong et al., 2025b) and effectiveness in RAG (Wang et al., 2025a), cross-domain

*Table 3.* **Main results on HolisQA across five domains.** We report **F1** (answer overlap), **CR** (Context Recall: how much gold context is covered by retrieved evidence), and **AR** (Answer Relevancy: evaluator-judged relevance of the answer to the question), all scaled to % for readability. **Bold** indicates best per column. NaiveGeneration has CR= 0 by definition (no retrieval).

| Baselines | Medicine | | | Computer Science | | | Business | | | Biology | | | Psychology | | |
|---|---|---|---|---|---|---|---|---|---|---|---|---|---|---|---|
| | F1 | CR | AR | F1 | CR | AR | F1 | CR | AR | F1 | CR | AR | F1 | CR | AR |
| *Naive Baselines* | | | | | | | | | | | | | | | |
| NaiveGeneration | 12.63 | 0.00 | 44.70 | 18.93 | 0.00 | 48.79 | 18.58 | 0.00 | 46.14 | 11.71 | 0.00 | 45.76 | 22.91 | 0.00 | 50.00 |
| BM25 | 17.72 | 52.04 | 50.64 | 24.00 | 39.12 | 52.40 | 28.11 | 37.06 | 55.52 | 19.61 | 43.02 | 52.32 | 30.46 | 33.44 | 56.63 |
| StandardRAG | 26.87 | 61.08 | 56.24 | 28.87 | 49.44 | 57.10 | 47.57 | 46.79 | 67.42 | 28.31 | 42.69 | 57.58 | 37.19 | 52.21 | 59.85 |
| *Graph-based RAG* | | | | | | | | | | | | | | | |
| GraphRAG Global | 17.13 | 54.56 | 48.19 | 23.75 | 37.65 | 53.17 | 23.62 | 25.01 | 48.12 | 20.67 | 40.90 | 52.41 | 31.09 | 34.26 | 54.62 |
| GraphRAG Local | 19.03 | 56.07 | 49.52 | 25.10 | 39.90 | 53.30 | 25.01 | 27.36 | 49.05 | 22.21 | 41.88 | 52.73 | 32.31 | 35.22 | 55.02 |
| LightRAG | 12.16 | 52.38 | 44.15 | 22.59 | 41.86 | 51.62 | 29.98 | 34.22 | 54.50 | 17.70 | 41.24 | 50.32 | 33.63 | 45.54 | 56.42 |
| *Structural / Causal Augmented* | | | | | | | | | | | | | | | |
| HippoRAG2 | 21.12 | 57.50 | 51.08 | 16.94 | 21.05 | 47.29 | 21.10 | 18.34 | 45.83 | 12.60 | 16.85 | 44.56 | 20.10 | 34.13 | 46.77 |
| LeanRAG | 34.25 | 60.43 | 56.60 | 30.51 | 57.61 | 55.45 | 48.30 | 59.29 | 60.35 | 33.82 | 58.43 | 56.10 | 42.85 | 57.46 | 58.65 |
| CausalRAG | 31.12 | 58.90 | 58.77 | 30.98 | 54.10 | 57.54 | 45.20 | 44.55 | 66.10 | 33.50 | 51.20 | 58.90 | 42.80 | 55.60 | 61.90 |
| CausalRAG2 (ours) | **36.45** | **69.91** | **60.65** | **31.60** | **60.94** | **58.34** | **51.51** | **67.34** | **68.76** | **34.80** | **61.97** | **59.99** | **44.42** | **60.87** | **63.53** |

*Table 4.* **Main results on five QA datasets.** Metrics follow Table 3: **F1**, **CR** (Context Recall), and **AR** (Answer Relevancy), reported in %. **Bold** and underline denote best and second-best per column.

| Baselines | MSMARCO | | | NQ | | | TwoWiki | | | QASC | | | HotpotQA | | |
|---|---|---|---|---|---|---|---|---|---|---|---|---|---|---|---|
| | F1 | CR | AR | F1 | CR | AR | F1 | CR | AR | F1 | CR | AR | F1 | CR | AR |
| *Naive Baselines* | | | | | | | | | | | | | | | |
| NaiveGeneration | 5.28 | 0.00 | 15.06 | 7.17 | 0.00 | 10.94 | 9.15 | 0.00 | 11.77 | 2.69 | 0.00 | 13.74 | 14.38 | 0.00 | 15.74 |
| BM25 | 6.97 | 45.78 | 20.33 | 4.68 | 49.98 | 9.13 | 9.43 | 37.12 | 13.73 | 2.49 | 6.12 | 13.17 | 15.81 | 41.08 | 16.08 |
| StandardRAG | 14.93 | 48.55 | 31.11 | 7.57 | 45.82 | 11.14 | 10.33 | 32.28 | 13.57 | 2.01 | 5.50 | 13.16 | 6.68 | 43.17 | 14.66 |
| *Graph-based RAG* | | | | | | | | | | | | | | | |
| GraphRAG Global | 9.41 | 3.65 | 13.08 | 3.91 | 4.48 | 8.00 | 1.41 | 9.42 | 9.55 | 0.68 | 3.38 | 3.56 | 6.28 | 14.59 | 16.26 |
| GraphRAG Local | 30.87 | 25.71 | 57.76 | 23.56 | 44.56 | 44.68 | 18.85 | 32.03 | 37.29 | 8.30 | 9.54 | 46.59 | 33.14 | 44.07 | 40.82 |
| LightRAG | 37.70 | 54.22 | 63.54 | 24.97 | 60.65 | 50.53 | 14.44 | 40.98 | 36.56 | 8.20 | 20.40 | 44.35 | 28.39 | **48.17** | 43.78 |
| *Structural / Causal Augmented* | | | | | | | | | | | | | | | |
| HippoRAG2 | 23.35 | 45.45 | 55.18 | 29.64 | 57.21 | 37.50 | 18.47 | **55.53** | 17.34 | **14.73** | 4.38 | **49.94** | 38.80 | 42.06 | 24.66 |
| LeanRAG | 38.02 | 54.01 | 58.49 | 35.46 | 65.91 | 49.87 | 20.27 | 40.53 | 38.37 | 13.19 | 22.80 | 45.51 | 48.68 | 46.29 | 43.50 |
| CausalRAG | 27.66 | 39.38 | 46.03 | 29.45 | 68.04 | 17.35 | 15.93 | 28.38 | 19.76 | 7.65 | 46.86 | 35.56 | 40.00 | 27.83 | 21.32 |
| CausalRAG2 (ours) | **38.40** | **60.48** | **66.02** | **49.50** | **70.36** | **55.09** | **31.97** | 41.95 | **42.67** | 13.35 | **70.80** | 49.40 | **64.83** | 40.30 | **45.72** |

experts further validated QA quality and the legitimacy of induced causal relations.

**Baselines.** We compare CausalRAG2 against eight baselines spanning three retrieval paradigms. First, to cover Naive and Flat approaches, we include **Naive Generation** (no retrieval) as a lower bound, alongside **BM25** (sparse) and **Standard RAG** (Lewis et al., 2020) (dense embedding-based), representing mainstream unstructured retrieval. Second, we evaluate established graph-based frameworks: **GraphRAG** (Local and Global) (Edge et al., 2025), utilizing community summaries; and **LightRAG** (Guo et al., 2025), relying on dual-level keyword-based search. Third, we benchmark against RAGs with structured or causal augmentation: **HippoRAG 2** (Gutiérrez et al., 2025), utilizing passage nodes and Personalized PageRank diffusion; **LeanRAG** (Zhang et al., 2026), employing semantic aggregation hierarchies and tree-based LCA re-

trieval; and **CausalRAG** (Wang et al., 2025a), which incorporates causality without explicit causal reasoning. This selection comprehensively covers the spectrum from unstructured search to advanced structure-aware and causally augmented graph methods.

**Metrics.** For metrics, we first report the token-level answer quality metric F1 for surface robustness. To measure whether retrieval actually supports generation, we additionally compute grounding metrics, context recall and answer relevancy (Es et al., 2025), which jointly capture coverage and answer quality (see Appendix F.4).

**Implementation Details.** For all experiments, we utilize gpt-5-nano as the backbone LLM for both the open IE extraction and generation stages, and Sentence-BERT (Reimers & Gurevych, 2019) for semantic vectorization. For CausalRAG2, we set the hierarchical seed budget to

$K_L = 3$ for modules and $K_0 = 3$ for entities. Causal gates are enabled by default except in the ablation study. Experiments run on a cluster using 10-way job arrays; each task uses 2 CPU cores and 16 GB RAM (20 cores, 160GB in total). See more implementation details in Appendix F.3.

## 5.2. Main Experiments

**Overall Performance (RQ1).** CausalRAG2 consistently achieves superior performance across all HolisQA domains and standard QA metrics (Table 3, Table 4). While traditional methods (e.g., BM25, Standard RAG) struggle with structural dependencies, graph-based baselines exhibit distinct limitations. GraphRAG Global relies heavily on high-level community summaries and largely suffers from detailed QA tasks, necessitating its GraphRAG Local variant to balance the granularity trade-off. LightRAG struggles to achieve competitive results, limited by its coarse-grained key-value lookup mechanism. Regarding structurally augmented methods, while LeanRAG (utilizing semantic aggregation) and HippoRAG2 (leveraging phrase/passage nodes) yield slight improvements in context recall, they fail to fully break information isolation compared to our causal gating mechanism. Finally, although CausalRAG occasionally attains high Answer Relevancy due to its causal reasoning capability, it struggles to scale to large datasets due to the lack of efficient knowledge graph organization.

**Holistic Comprehension vs. QA (RQ2).** The contrast between the results on HolisQA (Table 3) and standard QA datasets (Table 4) is revealing. On popular QA benchmarks, entity-centric methods like LightRAG, GraphRAG-Local, LeanRAG could occasionally achieve good scores. However, their performance degrades collectively and significantly on HolisQA. A striking counterexample is GraphRAG-Global: while its reliance on community summaries hindered performance on granular standard QA tasks, now it rebounds significantly in HolisQA. This discrepancy strongly suggests that standard QA datasets, which often favor short answers, **implicitly reward the entity-centric paradigm**. In contrast, HolisQA, with its open-ended questions and dense logical structures, **necessitates a comprehensive understanding** of the underlying document—a scenario closer to real-world applications. Notably, CausalRAG2 is the only framework that remains robust across this paradigm shift, demonstrating competitive performance on both entity-centric QA and holistic comprehension tasks.

**Reconciling the Accuracy-Grounding Trade-off (RQ3).** CausalRAG2 effectively reconciles the fundamental tension between Recall and Precision. While hierarchical causal gating expands traversal boundaries to secure superior **Context Recall (Globality)**, the explicit **causal path identification** rigorously prunes spurious noise to maintain high **F1 Score**

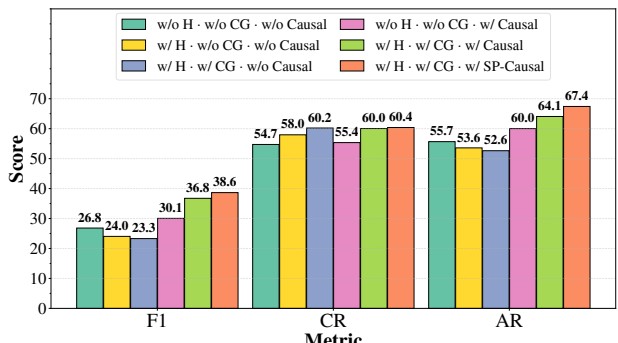

*Figure 3.* **Ablation Study.** H: Hierarchical Structure; CG: Causal Gates; Causal/SP-Causal: Standard vs. Spurious-Aware Causal Identification. w/o and w/ denote exclusion or inclusion.

**and Answer Relevancy (Locality)**. This dual mechanism allows CausalRAG2 to simultaneously optimize for global coverage and local groundedness, achieving a balance often missed by prior methods.

## 5.3. Ablation Study

To address **RQ4**, we ablate hierarchy, causal gates, and causal path refinement components (see Figure 3), finding that their combination yields optimal results. Specifically, adding hierarchy or causal gates alone broadens retrieval coverage and improves Context Recall, yet the additional cross-module evidence also introduces noise that depresses F1 and Answer Relevancy. The spurious-aware causal path refinement cleans this noise while preserving the expanded coverage, so all three metrics rise together only when structural expansion is paired with causal filtering. The full configuration thereby reconciles global coverage with local precision and outperforms every isolated component. Relative to CausalRAG (Wang et al., 2025a), a flat-causal baseline without hierarchical organization (Table 1), CausalRAG2 gains a mean +15.5 F1 across the five standard QA datasets. The gap scales with graph heterogeneity, reaching +24.83 F1 on HotpotQA (2,359 modules) and narrowing to +0.62 F1 on HolisQA-CompSci (158 modules), aligning with the prediction in Section 3 that hierarchy matters more when flat traversal cannot reliably reach distant evidence.

## 5.4. Scalability Analysis

**Robustness to Information Scale (RQ5).** To assess robustness against information overload, we evaluated performance across varying source text lengths ($5k$ to $1.5M$ characters) sampled from HolisQA, reporting the mean of F1, Context Recall, and Answer Relevancy (see Figure 4). As illustrated, CausalRAG2 (red line) exhibits remarkable stability across all scales, maintaining high scores even at 1.5M characters. This confirms that our hierarchical causal gating structure effectively encapsulates complexity, en-

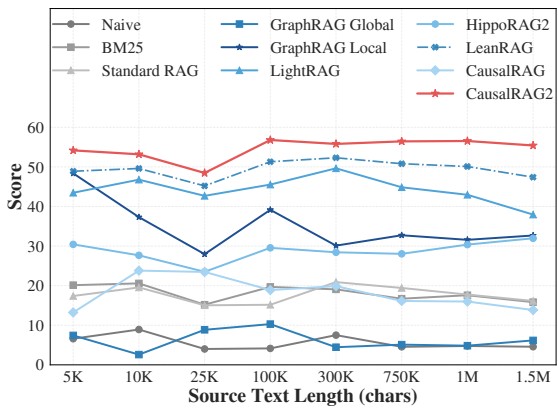

*Figure 4.* Scalability analysis of CausalRAG2 and other RAG baselines across varying source text lengths (5K to 1.5M characters).

abling the retrieval process to scale via causal gates without degrading reasoning fidelity.

## 5.5. Sensitivity Analysis

To assess the robustness of CausalRAG2 to its retrieval configuration, we sweep four key hyperparameters ($K_0$, $K_L$, $w_{\text{causal}}$, $\gamma$) governing seeding, edge weighting, and hop decay across a range of values on HolisQA-CompSci. Across all configurations, F1, Context Recall, and Answer Relevancy remain within narrow operating ranges, with no setting substantially degrading performance. As expected, enlarging the seed budget at either level expands coverage and lifts Context Recall, while exerting mild downward pressure on F1 as more peripheral evidence enters the retrieval pool. Overall, CausalRAG2 is not sensitive to the precise choice of these parameters within reasonable ranges. We further test the stability across different LLM backbones and random seeds, finding that larger-capacity backbones yield highly stable filtered subgraphs across repeated runs, while smaller models show greater variation. Full per-value hyperparameter results and per-backbone results are reported in Appendix H.

## 5.6. Cost Analysis

To assess the practicality of CausalRAG2, we measure both offline construction cost (per 1K corpus tokens) and online query cost across all baselines on HolisQA (**RQ6**). Table 5 summarizes the per-unit cost and time; the full breakdown including LLM token counts is provided in Appendix G.

CausalRAG2 incurs $0.0015 per 1K corpus tokens for offline graph construction, comparable to structural augmented RAGs, and roughly half the cost of GraphRAG. Construction time follows the same pattern, indicating that the LLM calls for causal gate verification do not impose disproportionate offline overhead because of our Top-Down Pruning algorithms (see Algorithm 2). Online query cost is $0.00049 per query with sub-second latency, on par with

graph-based baselines and well below GraphRAG-Global ($0.00068). The causal path filtering step does not introduce a noticeable query-time bottleneck. Combined with the retrieval quality gains reported in Tables 3 and 4, these results show that CausalRAG2 achieves its improvements within a computational budget comparable to existing graph-based RAG methods.

| Method | Construction (/1K) | | Query | |
| --- | --- | --- | --- | --- |
| | Cost ($) | Time (s) | Cost ($) | Time (s) |
| NaiveGeneration | 0.0000 | 0.000 | 0.000078 | 0.354 |
| BM25 | 0.0000 | 0.000 | 0.00011 | 0.562 |
| StandardRAG | 0.0000 | 0.000 | 0.00012 | 0.638 |
| GraphRAG-Global | 0.0034 | 19.694 | 0.00068 | 1.850 |
| GraphRAG-Local | 0.0034 | 19.694 | 0.00051 | 1.306 |
| LightRAG | 0.0015 | 7.317 | 0.00018 | 0.487 |
| HippoRAG2 | 0.00092 | 4.647 | 0.00032 | 0.641 |
| LeanRAG | 0.0012 | 5.589 | 0.00046 | 0.731 |
| CausalRAG | 0.0014 | 9.022 | 0.00052 | 0.849 |
| CausalRAG2 (ours) | 0.0015 | 9.622 | 0.00049 | 0.801 |

*Table 5.* Cost and latency comparison across baselines.

## 6. Conclusion

We presented CausalRAG2, a graph-based framework that addresses persistent issues, information isolation and entity-centric node matching, through hierarchical causal gating and explicit causal reasoning. Extensive experiments confirm its superior performance not only in standard QA but also in holistic comprehension, alongside robust scalability to large knowledge bases. We additionally introduced HolisQA to evaluate holistic comprehension across diverse domains. More broadly, the hierarchical causal gating mechanism offers a principled way to organize long contexts and long-term memory that LLMs or AI agents reason over. We hope our findings contribute to the ongoing development of related research.

## 7. Limitations

We note two limitations of CausalRAG2. First, the hierarchy depth is determined by the convergence of Leiden partitioning rather than optimized for the downstream task; learning task-adaptive hierarchies remains future work. Second, although HolisQA spans five academic domains, broader evaluation in domains such as legal or financial corpora would further strengthen claims about generalizability.

## Acknowledgements

This research was supported in part by NSF awards #2112606 and #2117439, and NAIRR #250362. We thank the anonymous reviewers for their constructive feedback.

## Impact Statement

This paper presents work whose goal is to advance the field of machine learning, specifically by improving the reliability and interpretability of retrieval-augmented generation. There are many potential societal consequences of our work, none of which we feel must be specifically highlighted here.

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

# A. Prompts used in Online Retrieval and Reasoning

This section details the prompt engineering employed during the online retrieval phase of CausalRAG2. We rely on Large Language Models to perform two critical reasoning tasks: identifying causal paths within the retrieved subgraph and generating the final grounded answer.

## A.1. Causal Path Identification

To address the *local spurious noise* issue, we design a prompt that instructs the LLM to act as a "causality analyst." The model receives a linearized list of potential evidence (nodes and edges) and must select the subset that forms a coherent causal chain.

**Spurious-Aware Selection (Main Setting).** Our primary prompt, illustrated in Figure 5, explicitly instructs the model to differentiate between valid causal supports (output in `precise`) and spurious associations (output in `ct_precise`). By forcing the model to articulate what is *not* causal (e.g., mere correlations or topical coincidence), we improve the precision of the selected evidence.

**Standard Selection (Ablation).** To verify the effectiveness of spurious differentiation, we also use a simplified prompt variant shown in Figure 6. This version only asks the model to identify valid causal items without explicitly labeling spurious ones.

## A.2. Final Answer Generation

Once the spurious-filtered support subgraph $S^\star$ is obtained, it is passed to the generation module. The prompt shown in Figure 7 is used to synthesize the final answer. Crucially, this prompt enforces strict grounding by instructing the model to rely *only* on the provided evidence context, minimizing hallucination.

---

```
---Role---
You are a careful causality analyst acting as a reranker for retrieval.

---Goal---
Given a query and a list of context items (short ID + content), select the most important items consisting  the causal graph and output them in "precise".
Also output the least important items as the spurious information in "ct_precise".
You MUST:
- Use only the provided items.
- Rank `precise` from most important to least important.
- Rank `ct_precise` from least important to more important.
- Output JSON only. Do not add markdown.
- Use the short IDs exactly as shown.
- Do NOT include any IDs in `p_answer`.

---Inputs---
Query:
{query}
Context Items (short ID | content):
{context_table}

---Output Format (JSON)---
{{
  "precise": ["C1", "N2", "E3"],
  "ct_precise": ["T7", "N9"],
  "p_answer": "concise draft answer"
}}

---Constraints---
- `precise` length: at most {max_precise_items} items.
- `ct_precise` length: at most {max_ct_precise_items} items.
- `p_answer` length: at most {max_answer_words} words.
```

*Figure 5.* Prompt for Causal Path Identification with **Spurious Distinction** (CausalRAG2 Main Setting). The model is explicitly instructed to segregate non-causal associations into a separate list to enhance reasoning precision.

---**Role---**
You are a careful causality analyst acting as a reranker for retrieval.

---**Goal---**
Given a query and a list of context items (short ID + content), select the most important items that best support answering the query as a **causal graph**.
You MUST:
- Use only the provided items.
- Rank the `precise` list from most important to least important.
- Output JSON only. Do not add markdown.
- Use the short IDs exactly as shown.
- Do NOT include any IDs in `p_answer`.
- If evidence is insufficient, say so in `p_answer` (e.g., "Unknown").

---**Inputs---**
Query:
{query}

Context Items (short ID | content):
{context_table}

---**Output Format (JSON)---**
{{
  "precise": ["C1", "N2", "E3"],
  "p_answer": "concise draft answer"
}}

---**Constraints---**
- `precise` length: at most {max_precise_items} items.
- `p_answer` length: at most {max_answer_words} words.

*Figure 6.* Ablation Prompt: Causal Path Identification **without** differentiating spurious relationships. This baseline is used to assess the contribution of the spurious filtering mechanism.

## B. Algorithm Details of CausalRAG2

This section provides granular details on the offline graph construction process and the specific algorithms used during the online retrieval phase, complementing the high-level description in Section 4.

**Function Definitions.** We clarify each function in Algorithm 1 as follows. LLM-ESTCAUSAL$(m_i, m_j)$ queries an LLM with modules $m_i$ and $m_j$ and returns a binary judgment indicating whether a plausible causal or logical dependency exists between them. GATEDTRAVERSAL$(U, \mathcal{H}, \mathcal{G}_c, h)$ executes a search from seed nodes $U$ over the graph comprising causal gates $\mathcal{G}_c$ within $h$ hops. CAUSALFILTER$(q, S_{\text{raw}})$ prompts an LLM (Appendix A.1) to select the most plausible causal reasoning subset $S^\star$ from retrieved subgraph $S_{\text{raw}}$ given query $q$, discarding spurious associations. LLM-GENERATE$(q, S^\star)$ generates the final answer conditioned on the filtered subgraph $S^\star$ (Appendix A.2).

### B.1. Graph Construction

**Entity Extraction and Deduplication.** The base graph $H_0$ is constructed by processing text chunks using LLM. We utilize the prompt shown in Figure 8, adapted from (Edge et al., 2025), to extract entities and relations. Since raw extractions from different chunks inevitably contain duplicates (e.g., "J. Biden" vs. "Joe Biden"), we employ a two-stage deduplication strategy. First, we perform surface-level canonicalization using fuzzy string matching. Second, we use embedding similarity to identify semantically identical nodes, merging their textual descriptions and pooling their supporting evidence edges.

**Hierarchical Partitioning.** We employ the Leiden algorithm (Traag et al., 2019) to maximize the modularity $Q$ of the partition. We recursively apply this partitioning to build bottom-up levels $H_1, \ldots, H_L$, stopping when the summary of a module fits within a single context window.

**Causal Gates.** The prompt we used to build causal gates is shown in Figure 9. Constructing causal gates via exhaustive pairwise verification across all modules results in a quadratic time complexity $O(N^2)$, where $N$ is the total number of modules. Consequently, as the hierarchy depth scales, this becomes computationally prohibitive for LLM-based verification.

---

---**Role**---
You are a helpful assistant answering the user's question.

---**Goal**---
Answer the question using the provided evidence context. A draft answer may be provided; use it only if it is supported by the evidence.

---**Evidence Context**---
{report_context}

---**Draft Answer (optional)**---
{draft_answer}

---**Question**---
{query}

---**Answer Format**---
Concise, direct, and neutral.

---

*Figure 7.* Prompt for Final Answer Generation. The model is conditioned solely on the filtered causal subgraph $S^\star$ to ensure groundedness.

To address this, we implement a **Top-Down Hierarchical Pruning** strategy that constructs gates layer-by-layer, from the coarsest semantic level ($H_L$) down to $H_1$. The strategy exploits the structure of the unified edge space $\mathcal{E}_{\text{uni}}$ (in Equation (2)): when two parent modules are already gate-connected, a path between their children exists via the parent gate plus hierarchical edges, so verifying additional child-pair gates yields no new traversal reachability (see full algorithm in Algorithm 2).

The pruning process follows three key rules:

1. **Layer-wise Traversal:** We iterate from top ($L$) (usually sparse) to bottom (1) (usually dense).

2. **Intra-layer Verification:** We first identify causal connections between modules within the current layer.

3. **Inter-layer Look-Ahead Pruning:** When searching for connections between a module $u$ (current layer) and modules in the next lower layer ($l - 1$), we prune the search space by:

   - Excluding $u$'s own children (handled by hierarchical inclusion).
   - **Excluding children of modules already gate-connected to** $u$, since the parent gate plus hierarchical edges already span $u$ to those children in $\mathcal{E}_{\text{uni}}$.

This strategy reduces construction complexity from quadratic to near-linear in practice while preserving traversal reachability over the unified edge space.

### B.2. Online Retrieval

**Hybrid Scoring and Diversity.**    To robustly anchor the query, our scoring function combines semantic and lexical signals:

$$s_\alpha(q, x) = \alpha \cdot \cos(\text{Enc}(q), \text{Enc}(x)) + (1 - \alpha) \cdot \text{Lex}(q, x), \tag{5}$$

where $\text{Lex}(q, x)$ computes the normalized token overlap between the query and the node's textual attributes (title and summary). We empirically set $\alpha = 0.7$ to favor semantic matching while retaining keyword sensitivity for rare entities. To ensure seed diversity, we apply Maximal Marginal Relevance (MMR) selection. Instead of simply taking the Top-$K$, we iteratively select seeds that maximize $s_\alpha$ while minimizing similarity to already selected seeds, ensuring the retrieval starts from complementary viewpoints.

**Edge Type Weights.**    In Equation (3), the weight function $w(\text{type}(e))$ controls the traversal behavior. We assign higher weights to Causal Gates ($w = 1.2$) and Hierarchical Links ($w = 1.0$) to encourage the model to leverage the organized structure, while assigning a lower weight to generic Structural Edges ($w = 0.8$) to suppress aimless local wandering.

---

**Algorithm 2** Top-Down Hierarchical Pruning for Causal Gates

---

**Require:** Hierarchy $\mathcal{H} = \{H_0, H_1, \ldots, H_L\}$
**Ensure:** Set of Causal Gates $\mathcal{G}_c$
 1: $\mathcal{G}_c \leftarrow \emptyset$
 2: **for** $l = L$ **down to** $1$ **do**
 3:    **for each** module $u \in H_l$ **do**
 4:       // 1. Intra-layer Verification
 5:       $ConnectedPeers \leftarrow \emptyset$
 6:       **for** $v \in H_l \setminus \{u\}$ **do**
 7:         **if** LLM_Verify$(u, v)$ **then**
 8:           $\mathcal{G}_c$.add$((u, v))$
 9:           $ConnectedPeers$.add$(v)$
10:         **end if**
11:       **end for**
12:       // 2. Inter-layer Pruning (Look-Ahead)
13:       **if** $l > 1$ **then**
14:         $Candidates \leftarrow H_{l-1}$
15:         // Prune own children
16:         $Candidates \leftarrow Candidates \setminus Children(u)$
17:         // Prune children of connected parents
18:         **for** $v \in ConnectedPeers$ **do**
19:           $Candidates \leftarrow Candidates \setminus Children(v)$
20:         **end for**
21:         // Only verify remaining candidates
22:         **for** $w \in Candidates$ **do**
23:           **if** LLM_Verify$(u, w)$ **then**
24:             $\mathcal{G}_c$.add$((u, w))$
25:           **end if**
26:         **end for**
27:       **end if**
28:    **end for**
29: **end for**
30: **return** $\mathcal{G}_c$

---

## B.3. Causal Path Reasoning

**Graph Linearization Strategy.** To reason over the subgraph $S_{raw}$ within the LLM's context window, we employ a linearization strategy that compresses heterogeneous graph evidence into a token-efficient format. Each evidence item $x \in S_{raw}$ is mapped to a unique short identifier $ID(x)$. The LLM is provided with a compact list mapping these IDs to their textual content (e.g., "N1: [Entity Description]"). This allows the model to perform selection by outputting a sequence of valid identifiers (e.g., "["N1", "R3", "N5"]"), minimizing token overhead.

**Spurious-Aware Prompting.** To mitigate noise, we design two variants of the selection prompt (in Appendix A.1):

- **Standard Selection:** The model is asked to output only the IDs of valid causal paths.

- **Spurious-Aware Selection (Ours):** The model is explicitly instructed to differentiate valid causal links from spurious associations (e.g., coincidental co-occurrence). By forcing the model to articulate (or internally tag) what is *not* causal, this strategy improves the precision of the final output list $S^\star$.

In both cases, the output is directly parsed as the final set of evidence IDs to be retained for generation.

**Broader connections to trustworthy and efficient AI systems.** CausalRAG2 is also connected to a broader line of work on building reliable, verifiable, and deployment-aware AI systems. Recent neurosymbolic and formal-verification

-Goal-
Given a text document that is potentially relevant to this activity and a list of entity types, identify all entities of those types from the text and all relationships among the identified entities.

-Steps-
**1. Identify all entities. For each identified entity, extract the following information:**
- entity_name: Name of the entity, capitalized
- entity_type: One of the following types: [{entity_types}]
- entity_description: Comprehensive description of the entity's attributes and activities
Format each entity as ("entity"{tuple_delimiter}<entity_name>{tuple_delimiter}<entity_type>{tuple_delimiter}<entity_description>)

**2. From the entities identified in step 1, identify all pairs of (source_entity, target_entity) that are *clearly related* to each other.**
For each pair of related entities, extract the following information:
- source_entity: name of the source entity, as identified in step 1
- target_entity: name of the target entity, as identified in step 1
- relationship_description: explanation as to why you think the source entity and the target entity are related to each other
- relationship_strength: a numeric score indicating strength of the relationship between the source entity and target entity
 Format each relationship as ("relationship"{tuple_delimiter}<source_entity>{tuple_delimiter}<target_entity>{tuple_delimiter}<relationship_description>{tuple_delimiter}<relationship_strength>)

**3. Return output in English as a single list of all the entities and relationships identified in steps 1 and 2. Use **{record_delimiter}** as the list delimiter.**

**4. When finished, output {completion_delimiter}**

######################
-Examples-
Example 1:
Entity_types: ORGANIZATION,PERSON
Text:
The Verdantis's C.................
Output:
("entity"{tuple_delimiter}CENTRAL INSTITUTION{tuple_delimiter}ORGANIZATION{tuple_delimiter}The Central Institution is the Federal Reserve of Verdantis, which..................
Example 2: .....
Example 3: .....

######################
-Real Data-
**Entity_types: {entity_types}**
**Text: {input_text}**
######################
**Output:**

*Figure 8.* Prompt for LLM-based Information Extraction (modified from GraphRAG (Edge et al., 2025)). Used in Step 1 of Offline Construction.

approaches study how model outputs can be audited, refined, or selectively trusted, including solver-backed verification for clinical vision-language reasoning (Singh et al., 2026d), verification-guided refinement for LLM reasoning (Singh et al., 2026a), and uncertainty modeling and uncertainty in treatment effect estimation for LLM-generated artifacts (Ganguly et al., 2026b; Singh et al., 2026b). Complementary work on model safety and controllable reasoning studies typicality-based detection of unsafe or out-of-distribution prompts (Ganguly et al., 2026a; 2025; Singh et al., 2026c) and token-level control of reasoning budgets (Yang et al., 2026; Wang et al., 2026a). Finally, systems-oriented work on efficient LLM training performance modeling (Zhang et al., 2025) and online anomaly detection via typicality learning (Chen et al., 2025) highlights the importance of scalability, efficiency, and robustness in practical AI deployments. Together, these efforts motivate our view that retrieval-augmented systems should move beyond surface-level semantic matching toward structured evidence, explicit reasoning support, and scalable reliability mechanisms.

## C. Visualization of CausalRAG2's Hierarchical Knowledge Graph

To provide an intuitive demonstration of CausalRAG2's structural advantages, we present 3D visualizations of the constructed knowledge graphs for two datasets: **HotpotQA** (see Figure 11) and **HolisQA-Biology** (see Figure 10). In these visualizations, nodes and modules are arranged in vertical hierarchical layers. The base layer ($H_0$), consisting of fine-grained entity nodes,

```
-Goal-
Given two text snippets A and B, decide whether there is any plausible causal relationship between them (either direction) under some reasonable context.

-Steps-
  1. Read A and B, and consider whether one could plausibly influence the other (directly or indirectly).
  2. Require a plausible mechanism; ignore mere correlation or co-occurrence.
  3. If uncertain or only associative, choose "no".

-Output-
Return exactly one token: "yes" or "no". No extra text.

#####################
-Real Data-
A: {a_text}
B: {b_text}
#####################
Output:
```

*Figure 9.* Prompt for Binary Causal Gate Verification. Used to determine the existence of causal links between module summaries.

is depicted in **grey**. The higher-level semantic modules ($H_1$ to $H_4$) are colored by their respective hierarchy levels. Crucially, the **Causal Gates**—which bridge topologically distant modules—are rendered as **red links**. To ensure visual clarity and prevent edge occlusion in this dense representation, we downsampled the causal gates, displaying only a representative subset ($r = 0.2$).

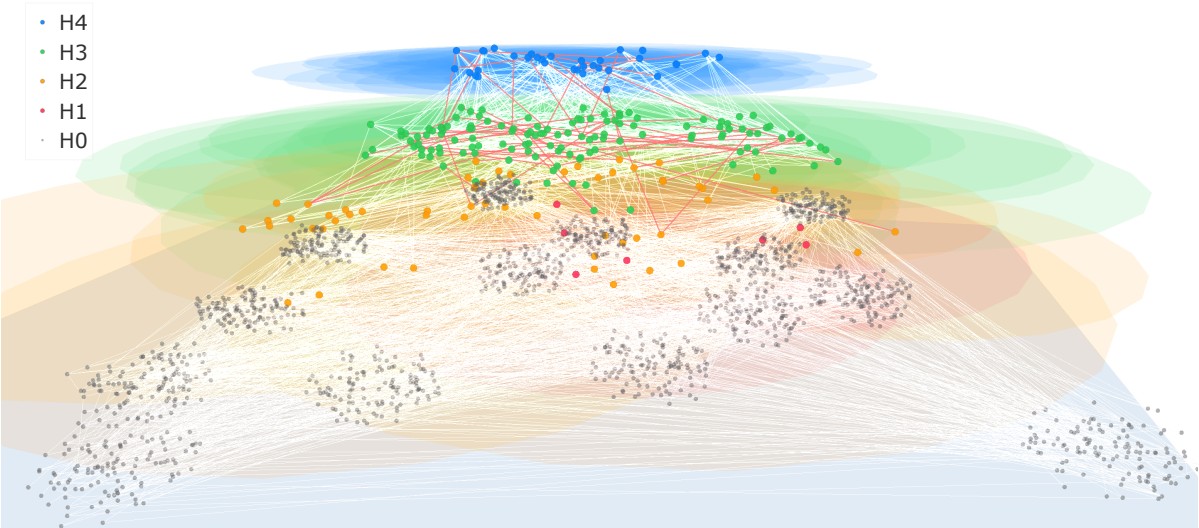

*Figure 10.* A **3D view** of the **Hierarchical Graph with Causal Gates** constructed from HolisQA-biology dataset.

## D. Case Studies

To illustrate how CausalRAG2 operates in practice, we present three real case studies drawn from HolisQA. Section D.1 provides a complete step-by-step pipeline trace with prompts and intermediate outputs. Section D.2 presents two additional qualitative examples focusing on how causal gates bridge topologically isolated modules to recover evidence that all baselines miss, and discusses robustness to incorrectly inferred gates.

### D.1. Complete Pipeline Trace

To concretely illustrate the CausalRAG2 full pipeline, we present a step-by-step execution trace on a query from the **HolisQA-Biology** dataset in Figure 12. The query asks for a comparison of specific enzyme activities (Apase vs. Pti-interacting kinase) in oil palm genotypes under phosphorus limitation, a task requiring the holistic comprehension of biology

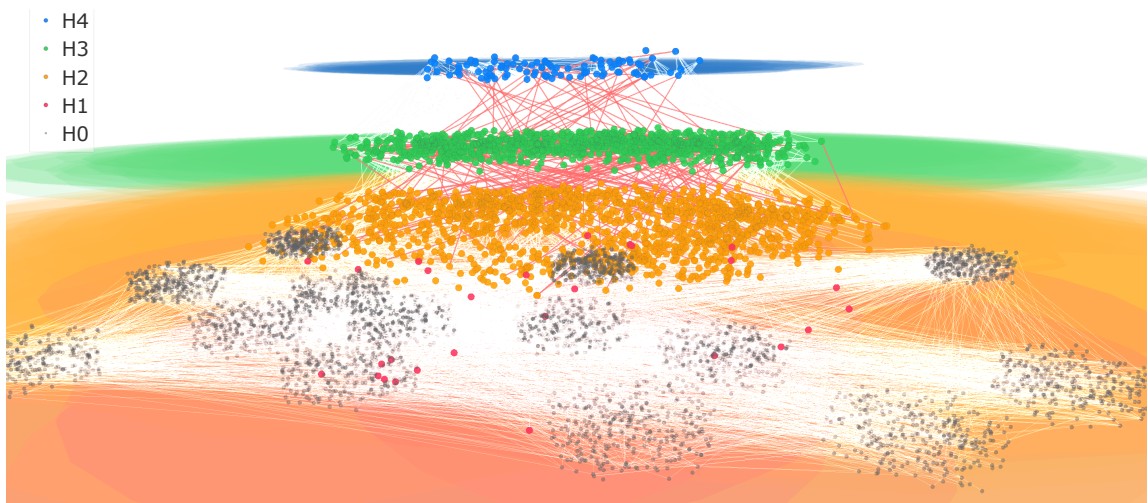

*Figure 11.* A **3D view** of the **Hierarchical Graph with Causal Gates** constructed from HotpotQA dataset.

knowledge in HolisQA dataset.

### D.2. Cross-Module Causal Bridging

We highlight two additional cases in HolisQA that illustrate how causal gates enable retrieval across topologically isolated modules.

**Case 1: Cross-Cluster Financial Reasoning.** *Query.* Which factors significantly impact firm value in energy-sector manufacturers, and how does profitability moderate these effects?

The gold evidence (a specific Indonesian manufacturing finance study) resides in an isolated module far from the seed communities about energy policy and corporate governance. Causal gates bridge from regulatory-economic modules to Southeast Asian corporate finance clusters, reaching the key text unit. All other baselines returned no relevant retrieval.

**Case 2: Cross-Domain Policy Reasoning.** *Query.* What is the primary purpose of research on AI-driven cancer diagnostics in LMICs, and what governance challenges are identified?

Answering requires connecting health governance, AI policy, and LMIC development, which reside in separate modules. CausalRAG2 seeds from health governance and AI readiness communities; causal gates then bridge to digital governance and LMIC-specific modules, reaching two complementary text units (one on AI in cancer diagnostics, one on broader AI societal challenges). Causal gates enable retrieval of both text units, while all other baselines failed.

## E. Experiments on the Effectiveness of Causal Gates

To validate the design of causal gates in CausalRAG2, we conduct two complementary sets of experiments. The first compares causal gates against alternative gate construction strategies (random, semantic, and perturbed variants) to verify that performance gains stem from causality rather than generic cross-module connectivity (Section E.1). The second tests retrieval behavior with causal gates enabled versus disabled, directly measuring the contribution of the gating mechanism itself (Section E.2).

### E.1. Comparison with Alternative Gate Types

To isolate the contribution of causality from generic cross-module shortcut effects, we compare five gate construction strategies on 500 randomly sampled QA pairs from HolisQA-CompSci, with the gate count held constant across Causal/Random/Semantic for fairness: **Causal** (ours, via LLM-ESTCAUSAL); **Random** (uniformly sampled module pairs); **Semantic** (top-cosine-similarity module pairs); **FP +25%** (Causal augmented with 25% additional random gates as false positives); **FN −25%** (Causal with 25% causal gates randomly removed as false negatives).

**Query:**
 How does the activity of acid phosphatase (Apase) and Pti-interacting serine/threonine kinase differ in oil palm genotypes under phosphorus limitation, and what are the implications for their adaptability?

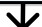

**Seed Stage:**
seed (matched via short_id_map): [T2, T4, T6, SP, CAT, ES, ADA, INDONESIA....]
e.g.:
- T2: [text_unit, score=0.4615] ols of PE direction and intensity, context-dependent microbial strategies, and the scarcity of long-term C balance assessments.........
- T4: [text_unit, score=0.4615] activity in P-optimum was higher than starvation and deficiency in leaf and root tissues in both genotypes, whereas Pti serine/t.......

**Post n-hop Subgraph:**
top_subgraph_nodes (by combined score). e.g.:
- E:dce66303-2b2c-472f-a964-da0b5529817d | SP (combined=0.4127)
- E:345eb0d6-56fb-4878-a05c-99f01d528cd8 | CAT (combined=0.3832)
sample_subgraph_edges. e.g.: ('up' means lower level module to higher level module.)
- struct:7->69:up | Xixia County Cornus Officinalis Pathogen Study in Henan -> Hebei-Liaoning-Yunnan Research Corridor (structural)
- struct:7->69 | Hebei-Liaoning-Yunnan Research Corridor -> Xixia County Cornus Officinalis Pathogen Study in Henan (structural)
- struct:7->68:up | Jehol Biota & Yixian Formation in NE China -> Hebei-Liaoning-Yunnan Research Corridor (structural)

**Causal LLM output:**
Identifying causal graph and spurious information (Full prompt available in appendix):
{
 "precise": ["T1", "T4", .........],
 "ct_precise": ["T7", "N9", ........],
 "p_answer": "Under phosphorus limitation in oil palm, Apase activity is expected to be induced (P deprivation), ..................."
}

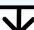

**Answer LLM output:**
Generate final answer (Full prompt available in appendix):
- Acid phosphatase (Apase): In oil palm, Apase activity in leaves and roots is higher under P optimum than under starvation/deficiency in both genotypes. This indicates Apase is not upregulated by phosphorus limitation in the studied conditions, suggesting it may not be the main adaptive response to P deprivation here.
- Pti-interacting serine/threonine kinase: Activity is higher in prolific versus non-prolific genotypes specifically under P-deficient conditions, indicating................

**Gold Answer:**
Apase activity was higher in P-optimal conditions than in starvation or deficiency for both leaf and root tissues across both oil palm genotypes. Conversely, Pti serine/threonine kinase activity was higher in prolific genotypes compared to non-prolific ones under P-deficient dosage. Additionally, abscisic acid content was higher in prolific genotypes during starvation and deficiency. These findings suggest that the prolific genotype is more adaptable to phosphorus deficiency, potentially..................

*Figure 12.* **A real example of CausalRAG2 on a biology-related query.** The diagram visualizes the data flow from initial seed matching and hierarchical graph expansion to the causal reasoning stage, where the model explicitly filters spurious nodes to produce a grounded, high-fidelity answer.

Our experiment shows that causal gates substantially outperform both Random and Semantic gates across all metrics (see Table 6), confirming that gains originate from causal structure rather than arbitrary cross-module connectivity. The monotonic degradation under FP and FN perturbations further indicates that retrieval quality scales with the precision of the gates.

**Expert Validation of Causal Gates.** Beyond performance comparison, two PhD-level domain experts independently evaluated 200 randomly sampled LLM-generated causal gates. The annotators confirmed 191/200 (95.5%) as valid causal relationships, with 92% inter-annotator agreement, supporting the reliability of our gating procedure.

## E.2. Effectiveness of Causal Gates: Enabled vs. Disabled

To isolate the real effectiveness of the causal gate in CausalRAG2, we conduct a controlled A/B test comparing gold context access with the gate disabled (*off*) versus enabled (*on*). The evaluation is performed on two datasets: **NQ** (Standard QA) and **HolisQA**. We define "Gold Nodes" as the graph nodes mapping to the gold context. Metrics are computed only on examples where gold nodes are mappable to the graph. While this section focuses on structural retrieval metrics, we evaluate the downstream impact of causal gates on final answer quality in our ablation study in Section 5.3.

| Gate Type | F1 | CR | AR |
|---|---|---|---|
| Random | 23.86 | 59.26 | 46.65 |
| Semantic | 24.87 | 59.37 | 47.14 |
| Causal | **31.62** | **60.98** | **58.31** |
| FP +25% | 27.87 | 60.36 | 52.65 |
| FN −25% | 28.60 | 57.26 | 54.23 |

*Table 6.* Comparison of gate variants on HolisQA.

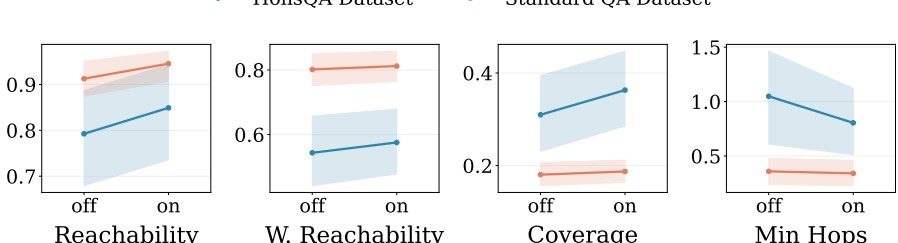

*Figure 13.* **Experiments on Causal Gate effectiveness.** We compare graph traversal performance with the causal gate disabled (*off*) versus enabled (*on*). Shaded areas represent 95% bootstrap confidence intervals. The causal gate significantly improves evidence accessibility (Reachability, Coverage) and traversal efficiency (lower Min Hops, higher Weighted Reachability).

**Metrics.** We report four structural metrics to evaluate retrieval quality and efficiency. Shaded regions in Figure 13 denote 95% bootstrap confidence intervals. **Reachability**: The fraction of examples where at least one gold node is retrieved in the subgraph. **Weighted Reachability (Depth-Weighted)**: A distance-sensitive metric defined as $\mathrm{DWR} = \frac{1}{1+\mathrm{min\_hops}}$ (0 if unreachable), rewarding retrieval at smaller graph distances. **Coverage**: The average proportion of total gold nodes retrieved per example. **Min Hops**: The mean shortest path length to gold nodes, computed on examples reachable in both *off* and *on* settings.

As shown in Figure 13, enabling the causal gate yields distinct behaviors across datasets. On the more complex HolisQA dataset, the gate provides a statistically significant improvement in reachability and coverage. This confirms that causal edges effectively bridge structural gaps in the graph that are otherwise traversed inefficiently. The increase in Weighted Reachability and decrease in min hops indicate that the gate not only finds *more* evidence but creates structural **shortcuts**, allowing the retrieval process to access evidence at shallower depths.

## F. Evaluation Details

### F.1. Detailed Graph Statistics

We provide the complete statistics for all knowledge graphs constructed in our experiments. Table 7 details the graph structures for the five standard QA datasets, while Table 8 covers the five scientific domains within the HolisQA dataset.

*Table 7.* **Graph Statistics for Standard QA Datasets.** Detailed breakdown of nodes, edges, and hierarchical module distribution.

| Dataset | Nodes | Edges | L3 | L2 | L1 | L0 | Modules | Domain | Chars |
|---|---|---|---|---|---|---|---|---|---|
| HotpotQA | 20,354 | 15,789 | 27 | 1,344 | 891 | 97 | 2,359 | Wikipedia | 2,855,481 |
| MS MARCO | 3,403 | 3,107 | 2 | 159 | 230 | 55 | 446 | Web | 1,557,990 |
| NQ | 5,579 | 4,349 | 2 | 209 | 244 | 50 | 505 | Wikipedia | 767,509 |
| QASC | 77 | 39 | - | - | - | 4 | 4 | Science | 58,455 |
| 2WikiMultiHop | 10,995 | 8,489 | 8 | 461 | 541 | 78 | 1,088 | Wikipedia | 1,756,619 |

### F.2. HolisQA Dataset

We introduce **HolisQA**, a comprehensive dataset designed to evaluate the holistic comprehension capabilities of RAG systems, explicitly addressing the entity-centric bias prevalent in existing QA datasets—where retrieving a single entity (e.g., a year or name) is often sufficient. Our goal is to enforce *holistic comprehension*, compelling models to synthesize coherent evidence from multi-sentence contexts.

*Table 8.* **Graph Statistics for HolisQA Datasets.** Graph structures constructed from dense academic papers across five scientific domains.

| Dataset | Nodes | Edges | L3 | L2 | L1 | L0 | Modules | Domain | Chars |
|---------|-------|-------|----|----|----|----|---------|--------|-------|
| HolisQA-Biology | 1,714 | 1,722 | - | 30 | 104 | 31 | 165 | Biology | 1,707,489 |
| HolisQA-Business | 2,169 | 2,392 | 8 | 77 | 166 | 41 | 292 | Business | 1,671,718 |
| HolisQA-CompSci | 1,670 | 1,667 | 7 | 28 | 91 | 30 | 158 | CompSci | 1,657,390 |
| HolisQA-Medicine | 1,930 | 2,124 | 7 | 56 | 129 | 34 | 226 | Medicine | 1,706,211 |
| HolisQA-Psychology | 2,019 | 1,990 | 5 | 45 | 126 | 35 | 211 | Psychology | 1,751,389 |

We collected high-quality scientific papers across multiple domains as our primary source (Priem et al., 2022), focusing exclusively on recent publications (2025) to minimize parametric memorization by the LLM. The dataset spans five distinct domains—Biology, Business, Computer Science, Medicine, and Psychology—to ensure domain robustness (see full statistics in Table 8). To necessitate cross-sentence reasoning, we avoid random sentence sampling; instead, we extract contiguous text *slices* from papers within each domain. Each slice is sufficiently long to encapsulate multiple interacting claims (e.g., Problem → Method → Result) yet short enough to remain self-contained, thereby preserving the logical coherence and contextual foundation required for complex reasoning. Subsequently, we employ a rigorous LLM-based generation pipeline to create Question-Answer-Context triples, imposing two strict constraints (as detailed in Figure 14):

1. **Integration Constraint:** The question must require integrating information from at least three distinct sentences. We explicitly reject trivia-style questions that can be answered by a single named entity (e.g., "Who founded X?").

2. **Evidence Verification:** The generation process must output the IDs of all supporting sentences. We validate the dataset via a necessity check, verifying that the correct answer cannot be derived if any of the cited sentences are removed.

Through this strict construction pipeline, HolisQA effectively evaluates the model's ability to perform holistic comprehension and isolate it from parametric knowledge, providing a cleaner signal for evaluating the effectiveness of structured retrieval mechanisms.

**Expert Validation of QA Quality.** To directly validate triple quality, two domain experts independently evaluated 200 randomly sampled (question, answer, context) triples from HolisQA-CompSci across three criteria: (i) whether the question is reasonable and answerable, (ii) whether the context sufficiently supports the answer, and (iii) whether the answer integrates multiple context sentences and is correct. The annotators confirmed $96.8\%$ of triples as fully valid on average, with $95.5\%$ (191/200) inter-annotator agreement.

---

**You are building a reading-comprehension dataset.**

You will receive a slice of sentences from a long document. Each line starts with a sentence ID, a tab, then the sentence text.

Generate {qas_per_run} question-answer pairs in JSON array format. Questions must require multi-sentence reasoning and an understanding of the overall slice. Avoid short factual questions, named-entity trivia, or single-sentence lookups.

**Each JSON item must include:**
- "question": string
- "answer": string (2-4 sentences)
- "context_sentence_ids": array of {min_context}-{max_context} IDs drawn only from the provided slice
  Return JSON only, no extra text.

**Sentences:**
{slice_text}

---

*Figure 14.* Prompt for generating the Holistic Comprehension Dataset (Question-Answer-Context Triplets) from academic papers.

### F.3. Implementation

We consistently use OpenAI's `gpt-5-nano` with a temperature of 0.0 to ensure deterministic generation. For vector embeddings, we employ the Sentence-BERT (Reimers & Gurevych, 2019) version of `all-MiniLM-L6-v2` with a dimensionality of 384. All evaluation metrics involving LLM-as-a-judge are implemented using the Ragas framework (Es et al., 2025), with `Gemini-2.5-Flash-Lite` serving as the underlying evaluation engine. To ensure a fair comparison

among all graph-based RAG methods, we utilize a unified root knowledge graph (see Appendix B.1 for construction details). For the retrieval stage, we set a consistent initial $k = 3$ across all baselines. All experiments were conducted on a high-performance computing cluster managed by Slurm. Each evaluation task was allocated uniform resources consisting of 2 CPU cores and 16 GB of RAM, utilizing 10-way job arrays for concurrent query processing.

### F.4. Grounding Metrics and Evaluation Prompts

We assess performance using two categories of metrics: (i) Lexical Overlap (F1 score), which measures surface-level similarity between model outputs and gold answers; and (ii) LLM-as-judge metrics, specifically Context Recall and Answer Relevancy, computed using a fixed evaluator model to ensure consistency (Es et al., 2025). To guarantee stable and fair comparisons across baselines with varying retrieval outputs, we impose a uniform cap on the retrieved context length and the number of items passed to the evaluator. The specific prompt template used for assessing Answer Relevancy is illustrated in Figure 15.

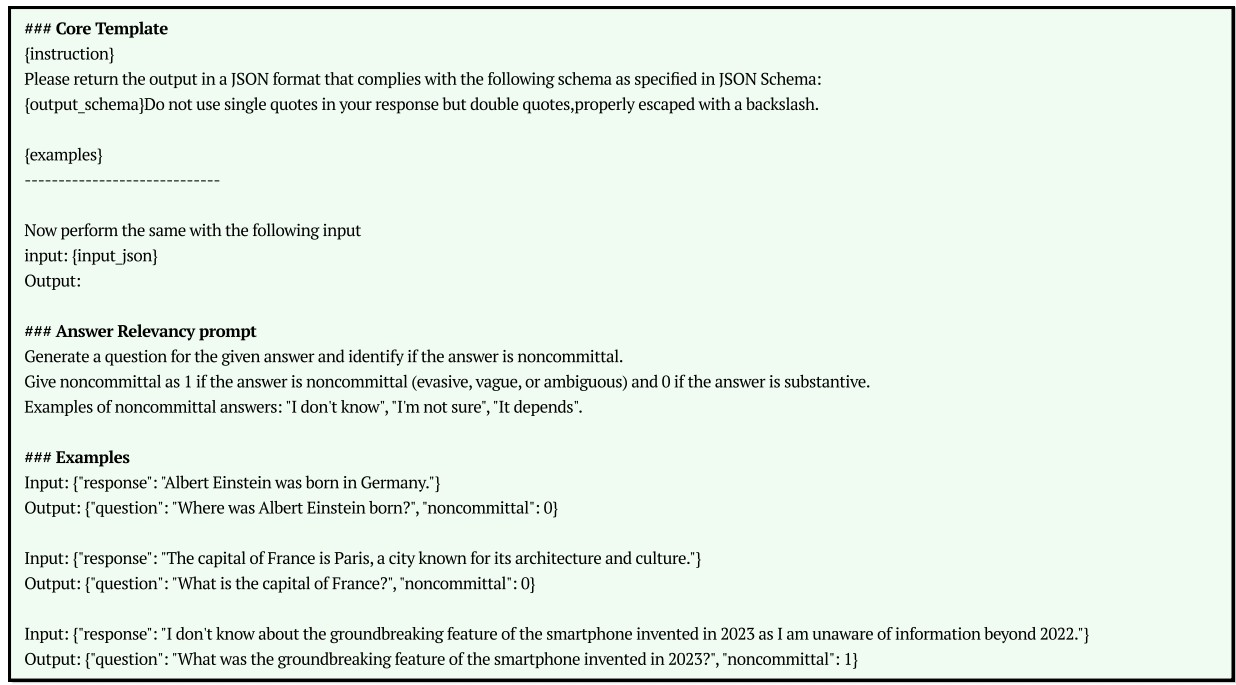

*Figure 15.* Example prompt used in RAGAS: Core Template and Answer Relevancy (Es et al., 2025).

# G. Full Cost and Latency Breakdown

Table 9 provides the complete breakdown of cost and latency across all baselines, including LLM token counts for both offline construction (normalized per 1K corpus tokens) and online query processing.

# H. Sensitivity Analysis: Full Results

We report the full results of two complementary sensitivity analyses: a sweep over CausalRAG2's retrieval hyperparameters (Section H.1), and a stability check of the causal filtering step across different LLM backbones and random seeds (Section H.2).

### H.1. Hyperparameter Sweeps

We sweep four key hyperparameters of CausalRAG2 across six values each on HolisQA-CompSci, with all other parameters held at their defaults ($K_0 = K_L = 3$, $w_{\text{causal}} = 1.2$, $\gamma = 0.7$). Table 10 reports F1, Context Recall (CR), and Answer Relevancy (AR) for each value.

| Method | Construction (per 1K corpus tokens) | | | Online Query (per query) | | |
|---|---|---|---|---|---|---|
| | Tokens | Time (s) | Cost ($) | Tokens | Time (s) | Cost ($) |
| NaiveGeneration | 0 | 0.000 | 0.0000 | 553.63 | 0.354 | 0.000078 |
| BM25 | 0 | 0.000 | 0.0000 | 967.90 | 0.562 | 0.00011 |
| StandardRAG | 0 | 0.000 | 0.0000 | 1016.56 | 0.638 | 0.00012 |
| GraphRAG-Global | 20241.49 | 19.694 | 0.0034 | 6566.85 | 1.850 | 0.00068 |
| GraphRAG-Local | 20241.49 | 19.694 | 0.0034 | 4968.42 | 1.306 | 0.00051 |
| LightRAG | 11245.55 | 7.317 | 0.0015 | 1517.70 | 0.487 | 0.00018 |
| HippoRAG2 | 5404.08 | 4.647 | 0.00092 | 3125.54 | 0.641 | 0.00032 |
| LeanRAG | 8893.11 | 5.589 | 0.0012 | 5031.30 | 0.731 | 0.00046 |
| CausalRAG | 9764.49 | 9.022 | 0.0014 | 5468.44 | 0.849 | 0.00052 |
| **CausalRAG2 (ours)** | 9864.49 | 9.622 | 0.0015 | 5075.65 | 0.801 | 0.00049 |

*Table 9.* Complete cost and latency breakdown across baselines. Construction costs are normalized per 1K corpus tokens; query costs are reported per individual query.

## H.2. LLM Backbone Stability

To assess the stability of the spurious-aware causal filtering step under different LLM backbones and random seeds, we run the filtering procedure three times for each backbone on a fixed set of queries from HolisQA-CompSci, and report the mean pairwise Jaccard similarity between the filtered subgraphs $S^\star$ across the three runs.

The per-backbone results (mean across three runs) are:

- GPT-oss-120B: $(0.888 + 0.813 + 0.854)/3 = 0.85$

- Qwen3-4B: $(0.934 + 0.974 + 0.969)/3 = 0.96$

- Llama-3.2-3B: $(0.496 + 0.750 + 0.340)/3 = 0.53$

Larger-capacity backbones (GPT-oss-120B, Qwen3-4B) yield highly stable filtering decisions, with Jaccard similarity at or above $0.85$. The smaller Llama-3.2-3B drops to $0.53$, indicating that reliable causal judgment depends on sufficient model capacity, which is consistent with prior observations in the causal discovery literature that LLM-based causal judgment scales with model capability.

| $K_0$ (bottom-level seed budget) | | | | | |
|---|---|---|---|---|---|
| Value | 1 | 3 | 6 | 10 | 30 | 100 |
| F1 | 29.72 | 31.63 | 31.23 | 30.27 | 30.09 | 27.24 |
| CR | 57.39 | 60.86 | 60.98 | 60.52 | 61.67 | 61.96 |
| AR | 57.98 | 58.34 | 58.30 | 57.02 | 57.36 | 56.01 |
| $K_L$ (module-level seed budget) | | | | | |
| Value | 1 | 3 | 6 | 10 | 30 | 100 |
| F1 | 29.31 | 31.26 | 31.74 | 30.79 | 30.87 | 29.30 |
| CR | 60.25 | 60.93 | 61.91 | 61.87 | 61.72 | 61.98 |
| AR | 56.50 | 59.71 | 59.34 | 56.65 | 55.21 | 55.26 |
| $w_{\text{causal}}$ (causal edge weight) | | | | | |
| Value | 0.6 | 0.8 | 1.0 | 1.2 | 1.4 | 1.6 |
| F1 | 30.22 | 31.60 | 30.75 | 31.12 | 30.57 | 29.02 |
| CR | 58.18 | 60.94 | 61.58 | 60.17 | 59.70 | 57.57 |
| AR | 57.49 | 58.34 | 58.41 | 58.03 | 56.42 | 56.62 |
| $\gamma$ (hop decay) | | | | | |
| Value | 0.4 | 0.5 | 0.6 | 0.7 | 0.8 | 0.9 |
| F1 | 32.53 | 33.09 | 32.66 | 31.54 | 30.62 | 28.60 |
| CR | 55.94 | 57.19 | 58.29 | 60.49 | 60.69 | 61.03 |
| AR | 59.24 | 60.04 | 59.70 | 59.27 | 58.36 | 56.87 |

*Table 10.* Hyperparameter sensitivity on HolisQA-CompSci. Each block sweeps one parameter while holding others at defaults.

