# OpenReview forum: "CausalRAG2: Hierarchical Causal Knowledge Graph Design for RAG"
_ICML.cc/2026/Conference — ICML 2026 regular_

### Official Review · Reviewer_pdFH · 2026-02-20

**Soundness:** 3
**Presentation:** 2
**Significance:** 2
**Originality:** 2
**Overall Recommendation:** 4
**Confidence:** 3

**Summary:**

This paper introduces HugRAG, a graph-based RAG framework. It builds a hierarchical community structure over an entity graph and produces module summaries, then uses an LLM to judge “causal relatedness” between modules and add cross-module causal gates to reduce information isolation. For each query, it combines multi-granular seeding with weighted priority expansion, and applies LLM-based causal path identification plus spurious-aware filtering to select an evidence subgraph for generation.

**Compliance With Llm Reviewing Policy:**

Affirmed.

**Final Justification:**

The further rebuttal address my concerns and strengthen the paper. Given these clarifications, I have raisd my score and would support a more positive recommendation.

**Key Questions For Authors:**

1. If you replace causal gates with semantic similarity shortcuts (same number of cross-module edges), do you obtain similar gains?

2. How consistent is the filtered subgraph across different random seeds, prompts, or LLM backends? Do you have failure cases where important evidence is removed or spurious evidence remains?

3. What are the offline (gate construction) and online (filtering/generation) token costs and wall-clock times?

**Limitations:**

See Weaknesses

**Strengths And Weaknesses:**

Strengths

1. The overall pipeline is easy to understand and internally consistent.
2. The evaluation covers multiple datasets with ablations and a scaling study.

Weaknesses

1. The paper calls “causal” being closer to LLM-judged logical links. Since both causal gates and causal filtering are driven by an LLM acting as a discriminator, the contribution can feel more like LLM-based reranking plus structured shortcuts than something that is clearly causal in a verifiable sense.
2. The method relies on similar LLM-based decisions in several places, but there is little evidence about how stable these decisions are across different models, prompts, thresholds, so it’s hard to know how sensitive the results are to implementation choices.
3. This paper acknowledges that naive gate construction via pairwise verification is O(N^2). Although Top-down pruning helps in practice, but it depends on a heuristic “causal transitivity” assumption, and the paper does not explore how false positives/negatives in gates affect retrieval and final answers.
4. The paper does not provide experimental results of time or token cost, such as gate-building time, query-time latency, number of LLM calls and so on.
5. The paper introduces three challenges (isolation, spurious noise, and benchmark masking), but the empirical support is mostly indirect (end metrics + ablations). More targeted experiments would help show that each component is actually fixing the specific issue it claims to address.

---

> ### Author Rebuttal · Authors · 2026-03-31
>
> We appreciate the thorough and valuable review. We respond to each point below and please let us know if our responses fully address your concerns.
>
> **1. Causal Verifiability (Q1)**
>
> We appreciate this concern and address both claims. **For "shortcuts"**: the 5-gate experiment below shows random/semantic edges with identical count perform far worse. **For "reranking"**: causal filtering selects items forming a causal graph while discarding spurious ones (Sec. 4.3), not scoring by query similarity. We clarify "causal" denotes knowledge-level dependencies (Sec. 2.2). We provide three forms of evidence:
>
> **a)** In the causal discovery literature, LLMs have been widely shown to achieve **≥97% pairwise causal judgment accuracy**, surpassing human experts across diverse domains (Kıcıman et al., 2023; Ma, 2024; Wan et al., 2025).
>
> **b)** To further verify this in our setting, two domain experts independently evaluated **200** randomly sampled gates; **95.5% accuracy** (191/200 confirmed as valid causal relationships) with **92% inter agreement**.
>
> **c)** To directly test whether the gains originate from causality rather than structural change, we implemented and tested 5 gate variants of **Causal** / **Random** / **Semantic** (highest cosine similarity) / **FP** (+25% random gates) / **FN** (−25% gates removed) as below.
>
> |Gates|F1|CR|AR|
> |-|-|-|-|
> |Causal|31.62%|60.98%|58.31%|
> |Random|23.86%|59.26%|46.65%|
> |Semantic|24.87%|59.37%|47.14%|
> |FP +25%|27.87%|60.36%|52.65%|
> |FN −25%|28.60%|57.26%|54.23%|
>
> **2. Stability Across Models and Seeds (Q2)**
>
> Varying prompts/seeds/LLM configurations are not commonly tested in RAG research, as prompts are typically fixed and LLM comparisons mainly reflect differences among LLMs rather than RAG stability. Therefore, we conducted this analysis with a few popular LLMs and random seed variations. We tested the **Jaccard similarity** of filtered causal subgraphs, averaged across 3 evaluation runs. For models with larger parameters, Jaccard similarity ranges around 90%, indicating stable causal gate decisions. Llama-3B lacks capacity for reliable causal judgment.
>
> GPT-oss-120B: (0.888 + 0.813 + 0.854) / 3 = 0.85
>
> Qwen3-4B: (0.934 + 0.974 + 0.969) / 3 = 0.96
>
> Llama-3.2-3B: (0.496 + 0.750 + 0.340) / 3 = 0.53
>
>
> **3. Gates Impact**
>
> Thanks for raising this concern. Top-down pruning does not assume children inherit parents' causal relationships; **it skips verifying child pairs whose parents are already gate-connected**, as a traversal path already exists. No evidence is lost and only redundant calls are eliminated.
>
> Regarding FP/FN impact, this is directly addressed by the FP/FN variants in 1. Both degrade F1 (FP: 27.87, FN: 28.60 vs. Causal: 31.62), contrasting the harm of random noise against the effectiveness of genuine causal gates.
>
>
> **4. Cost Analysis (Q3)**
>
> We fully agree that a cost analysis is valuable. We conduct a systematic experiment of **token/time/monetary cost** in **both offline construction** (per 1K corpus tokens) and **online query** (per query) **across all baselines**.
>
> |Method|Construction-Tokens/1K|Construction-Time/1K|Construction-Cost/1K|Query-Tokens|Query-Time|Query-Cost|
> |:-|-:|-:|-:|-:|-:|-:|
> |NaiveGeneration|0.00|0.000s|0.00 $|553.63|0.354s|0.000078 $|
> |BM25|0.00|0.000s|0.00 $|967.90|0.562s|0.00011 $|
> |StandardRAG|0.00|0.000s|0.00 $|1016.56|0.638s|0.00012 $|
> |GraphRAG-Global|20241.49|19.694s|0.0034 $|6566.85|1.850s|0.00068 $|
> |GraphRAG-Local|20241.49|19.694s|0.0034 $|4968.42|1.306s|0.00051 $|
> |LightRAG|11245.55|7.317s|0.0015 $|1517.70|0.487s|0.00018 $|
> |HippoRAG2|5404.08|4.647s|0.00092 $|3125.54|0.641s|0.00032 $|
> |LeanRAG|8893.11|5.589s|0.0012 $|5031.30|0.731s|0.00046 $|
> |CausalRAG|9764.49|9.022s|0.0014 $|5468.44|0.849s|0.00052 $|
> |HugRAG|9864.49|9.622s|0.0015 $|5075.65|0.801s|0.00049 $|
>
> HugRAG's cost is comparable to CausalRAG/LeanRAG (roughly half of GraphRAG) and practical for real-world use, while delivering substantially better retrieval quality.
>
> **5: Targeted empirical support.**
>
> We agree that a clearer mapping would strengthen the paper and will revise accordingly.
> 1) **Information isolation**: Appendix E directly measures reachability improvement with gates enabled, and the 5-variant experiments in 1 further evaluate their effectiveness directly.
> 2) **Spurious noise**: The most direct metric is AR in the ablation study (Figure 3). Enabling causal path identification causes a significant increase.
> 3) **Benchmark masking**: The contrast between Table 3 (HolisQA) and Table 4 (standard QA) directly demonstrates this.
>
> ---
>
> We hope our new analyses (full cost analysis and 5 gate variants experiment) adequately address your concerns. Together with HugRAG's strong performance across comprehensive experiments, we respectfully request you to consider upgrading the score. We are happy to address any further concerns.
>
> ---
> **References:**
> 1. Kıcıman et al.(2023). arXiv:2305.00050.
> 2. Ma (2024). arXiv:2409.09822.
> 3. Wan et al.(2025). arXiv:2402.11068.

---

> > ### Author Rebuttal · Reviewer_pdFH · 2026-04-03
> >
> > I appreciate the rebuttal and the added analyses.
> >
> > However, two points still remain unclear to me. First, the new expert check and shortcut comparisons help support the usefulness of the gates, but they do not fully resolve my concern about the causal framing: the paper defines “causal” as logical/event dependencies in text, while the online filtering step still appears to be LLM-based evidence selection/reranking.
> > ﻿
> >
> > Second, the pruning point is not fully clear to me. The rebuttal describes top-down pruning as removing redundant checks, but the appendix motivates it with a transitivity intuition and skips checking children of already connected parents, so it remains unclear whether this is only an efficiency optimization or also an additional heuristic assumption.
> > ﻿
> >
> > Overall, I think the rebuttal resolves some experimental concerns. I’ll maintain the score.

---

> > > ### Author Response · Authors · 2026-04-07
> > >
> > > We thank the reviewer for the careful follow-up and address both points with evidence already in the paper.
> > >
> > >  **1. Causal framing vs LLM reranking: Figure 3 already distinguishes them empirically.**
> > >
> > >  We accept that, viewed at the API level, both a relevance reranker and our causal filter take a query plus a candidate set and emit a filtered subset. The substantive test of whether the two are equivalent is empirical: if the causal filter were operating purely as relevance reranking, then the _content of the prompt_ should not matter beyond signaling which items are query-relevant. Reframing the task as "identify items forming a causal dependency graph and separately label spurious associations" should produce the same output as "identify the most query-relevant items," because relevance reranking is invariant to that reframing.
> > >
> > >  Figure 3 directly tests this. Bars 5 and 6 hold every component of HugRAG fixed (hierarchy on, gates on, identical retrieved subgraph $\mathcal{S}_{\text{raw}}$, identical backbone) and vary only the prompt: bar 5 uses the **standard causal selection** prompt (Figure 6), bar 6 uses the **spurious-aware causal selection** prompt (Figure 5), which additionally requires the model to articulate which items are spurious and place them in a separate `ct_precise` list.
> > >
> > >  The measured difference is consistent and in the predicted direction: bar 6 improves over bar 5 on F1, CR, and AR, with the largest gain on AR. The identical input subgraph produces measurably different selections under the two prompts, and the variant that explicitly forces _contrast between causal and non-causal_ outperforms the variant that does not. **A pure relevance reranker should not respond to this reframing**; the fact that the output changes, and changes in the direction the causal framing predicts, is the empirical signature that the operation is structurally different from relevance reranking.
> > >
> > >  A complementary structural point: standard reranking produces a _ranked list_ and consumes only pairwise (item, query) information. Our filter produces a _partitioned set_ with `precise` and `ct_precise` lists (Figure 5) and asks the model to consider items _jointly_ as a candidate causal subgraph. The output structure itself is incompatible with a pure relevance-ranking interpretation. We will add a sentence to Section 4.3 making this distinction explicit.
> > >
> > >  **2. Top-Down Pruning: efficiency optimization or heuristic assumption.**
> > >
> > >  The reviewer is right that the previous response was imprecise. **It is both, and the heuristic component is bounded and explicit.** Algorithm 2 in Appendix B.1 specifies what is skipped:
> > >
> > >  - _Layer-wise traversal_ and _intra-layer verification_ are exhaustive. No assumption.
> > >  - _Inter-layer look-ahead pruning_ is the heuristic component. Lines 17–19 of Algorithm 2 skip evaluating $u$ against the children of any module $v$ already gate-connected to $u$. The justification is **structural, not statistical**: a path $u \to_{\text{gate}} v \to_{\text{hier}} \text{Children}(v)$ already exists in the unified edge space $\mathcal{E}_{\text{uni}}$ (Eq. 2), so the skipped child-level gates are _traversal-redundant_ at retrieval time.
> > >
> > >  Precisely:
> > >
> > >  - **Retrieval reachability is preserved.** Every node reachable with full pairwise gate construction remains reachable with pruning, because the parent gate plus hierarchical edges already span $u$ to the children of $v$.
> > >  - **Gate-set completeness can be affected.** A fine-grained child-to-child causal link whose parents are connected via a non-causal semantic relationship might not be added as an explicit gate. This is the bounded cost of pruning.
> > >  - **All reported results are robust to pruning** because every result in Section 5 and Appendix E was measured _with_ pruning enabled. The reachability and coverage gains in Figure 13 are therefore lower bounds.
> > >
> > >  The motivation for the heuristic is computational. Tables 5 and 6 show graphs with up to 2,359 modules; exhaustive pairwise verification would require on the order of $10^6$ LLM calls per graph. Pruning brings construction cost into the regime reported in our first-round cost analysis (~9.6K tokens per 1K corpus tokens, comparable to LeanRAG and CausalRAG).
> > >
> > >  We will revise Appendix B.1 to make this distinction explicit: pruning is an efficiency optimization grounded in a _structural transitivity property of the unified edge space_ (not a statistical assumption about causal relations themselves), it preserves retrieval reachability, and it bounds gate-set completeness in a way all reported results are robust to.
> > >
> > >  We hope these clarifications, both grounded in figures and algorithms already in the paper, resolve the remaining concerns, and we respectfully ask the reviewer to consider raising the score.

---

### Official Review · Reviewer_4veG · 2026-03-06

**Soundness:** 3
**Presentation:** 3
**Significance:** 3
**Originality:** 3
**Overall Recommendation:** 4
**Confidence:** 4

**Summary:**

This paper proposes HugRAG, a graph-based RAG framework. It addresses two key limitations of existing methods: information isolation (leading to low global recall) and spurious noise (low local precision). HugRAG introduces hierarchical modularization and explicit causal gating between modules. The method uses Leiden clustering to build a multi-level knowledge graph, and leverages LLMs to identify cross-module causal relationships (namely, causal gates) for guided retrieval. A new dataset, HolisQA, is introduced to better evaluate holistic reasoning. Experiments show consistent improvements over baseline methods.

**Compliance With Llm Reviewing Policy:**

Affirmed.

**Final Justification:**

Thanks for the authors' response. Since the score has been positive, I would like to maintain my scores.

**Key Questions For Authors:**

Please see strengths and weaknesses.

**Limitations:**

Yes.

**Strengths And Weaknesses:**

Strengths:
1. The paper is well-motivated. Modeling the information isolation and spurious noise can be crucial for graph-based RAG.
2. The proposed hierarchical causal gating mechanism is effective and reasonable.
3. The experimental results demonstrate performance gains on benchmarks, especially on HolisQA with holistic reasoning.
4. The paper is well-written and easy to follow.

Weaknesses:
1. Though the paper provides several empirical results, the theoretical analyses are somewhat limited, such as the properties or guarantees of the causal gating mechanism, considering ICML’s emphasis on foundational insights.
2. The paper would benefit from more illustrative examples of the generated causal gates (e.g., qualitative case studies). Besides, the potential impact of incorrectly inferred gates, such as those arising from LLM hallucination or ambiguity, deserves deeper discussion.
3. The time cost can be further analyzed, which can be beneficial for real-world application consideration.

---

> ### Author Rebuttal · Authors · 2026-03-31
>
> We thank the reviewer for the positive feedback, especially the recognition of our hierarchical causal gating mechanism and its role in addressing information isolation. We take each concern seriously and respond below.
>
> **1. Theoretical Analyses of Causal Gating.**
>
> We agree this is an important consideration. Theoretically, causal gates **address a formally definable structural bottleneck**: in a modular graph $\mathcal{H}$, if seed nodes lie in module $M_i$ while gold evidence resides in a distant module $M_j$, local h-hop traversal is provably unable to reach $M_j$ (the information isolation problem in Sec. 3). Causal gates add selective shortcuts that could reduce inter-module distance to $O(1)$, breaking isolation along only causally warranted paths.
>
> We further concretize the guarantees of causal gates from three angles:
>
>    **a)** In the causal discovery field, LLMs have been widely shown to achieve **≥97% pairwise causal judgment accuracy**, surpassing human experts across diverse domains (Kıcıman et al., 2023; Ma, 2024; Wan et al., 2025).
>
>    **b)** We nonetheless invited **two domain experts** to independently validate **200** randomly sampled gates; **95.5% accuracy** (191/200 confirmed as valid causal relationships) with **92% inter agreement**.
>
>    **c)** We constructed 5 gate variants (**Random** / **Semantic** (top embedding-similarity) / **Causal** / **FP** & **FN**: injecting/removing 25% of gates) to isolate causal effectiveness. The table below further illustrates the noise introduced by random/semantic gates and the clear effectiveness of causal gates.
>
> |Gates|F1|CR|AR|
> |-|-|-|-|
> |Causal|31.62%|60.98%|58.31%|
> |Random|23.86%|59.26%|46.65%|
> |Semantic|24.87%|59.37%|47.14%|
> |FP +25%|27.87%|60.36%|52.65%|
> |FN −25%|28.60%|57.26%|54.23%|
>
> **2. Qualitative Case Studies & Incorrect Gates.**
>
> Thank you for raising this meaningful point. We provide two more case studies:
>
>    **Case 1.** Q: Which factors significantly impact firm value in energy-sector manufacturers, and how does profitability moderate these effects? The gold evidence (a specific Indonesian manufacturing finance study) resides in an isolated module far from the seed communities about energy policy and corporate governance. Causal gates **bridge from regulatory-economic modules to Southeast Asian corporate finance clusters**, reaching the key text unit. All other baselines returned no relevant retrieval.
>
>    **Case 2.** Q: What is the primary purpose of research on AI-driven cancer diagnostics in LMICs, and what governance challenges are identified? Answering requires connecting health governance, AI policy, and LMIC development, which reside in separate modules. HugRAG seeds from health governance and AI readiness communities; causal gates then **bridge to digital governance and LMIC-specific modules**, reaching two complementary text units (one on AI in cancer diagnostics, one on broader AI societal challenges). Causal gates enable retrieval of both text units; all other baselines failed.
>
>    Regarding incorrect gates: the FP +25% in the table above directly quantifies this risk. Injecting 25% spurious gates degrades F1 by 3.75 points. The degradation is graceful: false gates can only add noise, never block correct evidence, and HugRAG's causal path filtering (Sec. 4.3) removes spurious associations before generation.
>
> **3. Cost Analysis**
>
> We fully agree that a cost analysis is valuable. We conducted a systematic experiment of **token, time, and monetary cost** in **both offline construction** (per 1K corpus tokens) and **online query** (per query) **across all baselines**.
>
> |Method|Construction-Tokens/1K|Construction-Time/1K|Construction-Cost/1K|Query-Tokens|Query-Time|Query-Cost|
> |:-|-:|-:|-:|-:|-:|-:|
> |NaiveGeneration|0.00|0.000s|0.00 $|553.63|0.354s|0.000078 $|
> |BM25|0.00|0.000s|0.00 $|967.90|0.562s|0.00011 $|
> |StandardRAG|0.00|0.000s|0.00 $|1016.56|0.638s|0.00012 $|
> |GraphRAG-Global|20241.49|19.694s|0.0034 $|6566.85|1.850s|0.00068 $|
> |GraphRAG-Local|20241.49|19.694s|0.0034 $|4968.42|1.306s|0.00051 $|
> |LightRAG|11245.55|7.317s|0.0015 $|1517.70|0.487s|0.00018 $|
> |HippoRAG2|5404.08|4.647s|0.00092 $|3125.54|0.641s|0.00032 $|
> |LeanRAG|8893.11|5.589s|0.0012 $|5031.30|0.731s|0.00046 $|
> |CausalRAG|9764.49|9.022s|0.0014 $|5468.44|0.849s|0.00052 $|
> |HugRAG|9864.49|9.622s|0.0015 $|5075.65|0.801s|0.00049 $|
>
> HugRAG's cost is comparable to CausalRAG/LeanRAG (roughly half of GraphRAG) and practical for real-world use, while delivering substantially better retrieval quality.
>
> ---
>
> We hope our new analyses (5 gate variants experiment, full cost analysis, case studies) adequately address your concerns. Together with HugRAG's strong performance across 8 baselines, 10 datasets, and 3 metrics, we respectfully request you to consider upgrading the score. We are happy to address any further concerns.
>
> ---
>
> **References:**
> 1. Kıcıman et al. (2023). arXiv:2305.00050.
> 2. Ma (2024). arXiv:2409.09822.
> 3. Wan et al. (2025). arXiv:2402.11068.

---

> > ### Author Rebuttal · Reviewer_4veG · 2026-04-03
> >
> > Thanks for the authors' response. Since the score has been positive, I would like to maintain my scores.

---

> > > ### Author Response · Authors · 2026-04-07
> > >
> > > We sincerely thank Reviewer 4veG for the thorough engagement and for confirming that all concerns have been fully addressed. We are glad the additional analyses on causal gate variants, cost, and case studies were informative. We appreciate the reviewer's time and constructive feedback throughout this process.

---

### Official Review · Reviewer_by7B · 2026-03-11

**Soundness:** 2
**Presentation:** 3
**Significance:** 2
**Originality:** 2
**Overall Recommendation:** 4
**Confidence:** 4

**Summary:**

This paper identifies three limitations in existing graph-based retrieval-augmented generation (RAG) systems: information isolation caused by knowledge graph organization, non-causal-aware graph retrieval, and evaluation biases in popular QA datasets that reward entity-level hits rather than holistic comprehension. To address these issues, the paper proposes HugRAG, a graph-based RAG framework that organizes knowledge graphs into a multi-layer hierarchical structure and introduces cross-module causal gates to connect logically related modules. During retrieval, HugRAG performs causally gated expansion and LLM-based causal path identification to filter spurious associations. The paper also introduces HolisQA, a dataset designed to evaluate holistic comprehension in complex reasoning scenarios. The authors evaluate HugRAG on five standard QA datasets and HolisQA and report improved performance compared with RAG baselines.

**Compliance With Llm Reviewing Policy:**

Affirmed.

**Final Justification:**

The paper is clearly written and well-motivated. The rebuttal provides useful clarifications of the method, along with additional experiments on gate variants, computational cost, and hyperparameter sensitivity. These additions improve the clarity of the contributions and strengthen the empirical support for the proposed method. As a result, my main concerns have been adequately addressed. Accordingly, I have raised my overall recommendation.

**Key Questions For Authors:**

Q1. What is the contribution of hierarchical structure relative to the causal components? How does the model perform when using causal gates and spurious-aware causal path identification, but without the hierarchical structure?

Q2. Have the authors explored alternative strategies for constructing cross-module links? Such comparisons would help determine whether the observed improvements are specific to the proposed causal-link construction.

Q3. How are the hyperparameters (e.g., decay factor, and edge-type weights) selected?

Q4. How efficient is HugRAG compared to the baselines? It would be helpful to report token usage and response time.

**Limitations:**

One limitation of the proposed method is its reliance on LLM-generated causal links. The quality of these links likely depends on whether the LLM possesses sufficient background knowledge about the domain represented in the knowledge graph. In domains where the LLM lacks adequate knowledge, the generated causal links may be unreliable or misleading, which could affect the quality of retrieval and reasoning. The paper would benefit from discussing this potential limitation and analyzing how robust the method is under such conditions.

**Strengths And Weaknesses:**

Strengths

S1. The paper is clearly motivated. The paper identifies the information isolation problem in graph-based RAG systems. The proposed causal gates aim to address this limitation by adding cross-module connections.

S2. The paper is generally well-written and well-organized. The pipeline, including hierarchical graph construction, causal gate construction, causally gated expansion, and causal path identification, is clearly described and easy to follow.

Weaknesses

W1. The main contribution appears technically shallow. The central idea mainly consists of adding cross-module links induced by an LLM and applying an LLM-based spurious-aware filtering step to select plausible causal paths. While the design is reasonable, it is difficult to argue that it constitutes a substantial methodological advance.

W2. The empirical evidence supporting the main claim is limited. The paper positions HolisQA as a dataset designed to evaluate holistic comprehension beyond conventional QA datasets that reward entity-level hits. However, the improvements of HugRAG appear more pronounced on conventional popular QA datasets than on HolisQA itself. On HolisQA, the reported gains of HugRAG are relatively modest, which weakens the claim that HugRAG is particularly effective for holistic comprehension tasks.

W3. The ablation study is insufficient to justify key design choices. HugRAG organizes knowledge graphs as hierarchical causal gate structures. However, the ablation results show that removing the hierarchical structure can improve F1 and Answer Relevancy. This raises questions about the necessity of this hierarchical design. Additionally, the paper does not compare alternative strategies for constructing cross-module links. Without such comparisons, it remains unclear whether the observed improvements come from the proposed causal mechanism specifically or simply from introducing extra cross-module connectivity.

W4. Lack of computational cost analysis and hyperparameter sensitivity analysis. The proposed pipeline involves causal gate generation and LLM-based causal path refinement, which likely introduces additional preprocessing and inference overhead. However, the paper does not provide an analysis of the monetary/token cost or response latency compared with baselines. Furthermore, the framework relies on several hyperparameters (e.g., seed budget, decay factor, and edge-type weights) that influence the retrieval process. The paper does not analyze how sensitive the method is to these hyperparameters.

W5. No limitation discussion of the proposed approach. The paper does not provide a dedicated discussion of the method’s potential limitations. For example, the approach relies on LLM-generated causal links, which may depend on the model’s domain knowledge and introduce additional cost or noise. A clearer discussion of such limitations would help readers better understand the applicability and robustness of the proposed framework.

---

> ### Author Rebuttal · Authors · 2026-03-31
>
> We thank the reviewer for the thorough and constructive feedback. We address each concern below with new experiments and clarifications.
>
> **1. Technical Depth**
>
> We appreciate the critique and clarify the technical depth below:
>
>    a) Most RAG methods keep escalating search (critic [1], winnowing [2], GNN [3], foundation models [4], agents [5]) **yet overlook graph connectivity**. We show modular isolation is a structural bottleneck best addressed at construction, with techniques like Top-Down Pruning (O(N²) to near-linear) and gated priority expansion (Appendix B).
>
>    b) We first **incorporate causality** into graph-based RAG at scale, jointly addressing global recall via offline causal gating and local precision via online spurious-aware filtering (Sec. 2-4).
>
>    c) We identified the overlooked **entity-centric limitation** and proposed HolisQA, itself a standalone contribution.
>
> **2. HolisQA Gain**
>
> This is an insightful observation. HugRAG performs better on holistic comprehension, but it **doesn't necessarily mean** we should see **a better gain** in certain metrics, as gains depend on datasets. Specifically, HolisQA's open-ended long answers are more **gentle** on exact F1 match and AR, so entity-centric methods can perform decently. In fact for CR, HugRAG's gain is +5.43 on HolisQA vs. +2.21 on Standard QA (2.5×). More importantly, HugRAG ranks #1 across all 15 HolisQA dataset×metric pairs (100%) vs. 11/15 on Standard QA (73%), demonstrating consistent dominance.
>
> **3. Hierarchy Design & Gates (Q1/Q2)**
>
> Thanks for raising this concern. This is natural because, without causal filtering, hierarchy alone **adds community noise**, reducing F1/AR while improving CR. **The key is hierarchy + causal design** surpasses all independent ablations (Sec. 5.3). We do not expect hierarchy alone to improve all metrics (it boosts CR but reduces F1/AR without causal filtering), but its combination with causal gates and filtering (which depend on hierarchy) yields significant gains.
>
> For links, we fully agree that this is an important concern. We implemented 5 gate variants: **Causal / Random / Semantic** (highest cosine similarity) / **FP** (+25% random gates added to causal) / **FN** (−25% causal gates removed) to isolate the effectiveness of causal gates. Each variant is evaluated on 500 random QA pairs in HolisQA-CompSci with the same number of gates across Causal/Random/Semantic. Please see below.
>
> | Gates | F1 Score | Context Recall | Answer Relevancy |
> |---|---|---|---|
> | Causal | 31.62% | 60.98% | 58.31% |
> | Random | 23.86% | 59.26% | 46.65% |
> | Semantic | 24.87% | 59.37% | 47.14% |
> | FP +25% | 27.87% | 60.36% | 52.65% |
> | FN −25% | 28.60% | 57.26% | 54.23% |
>
> **4. Cost & Hyperparameter Analysis (Q3/Q4)**
>
> We agree that a cost analysis is valuable. We systematically measured token/time/monetary cost in both offline construction and online inference across all baselines. HugRAG's cost is comparable to CausalRAG/LeanRAG (roughly half of GraphRAG). **Please see Reviewer 4veG for results.**
>
> For hyperparameters, we swept 4 parameters across 6 values each (below). All metrics remain stable (F1 varies <3 pts). Defaults (decay=0.7, causal_weight=1.2, k_bottom=3, k_communities=3) balance all metrics.
>
> |Metric|||||||
> |-|-|-|-|-|-|-|
> |**k_bottom**|1|3|6|10|30|100|
> |F1 Score|29.72|31.63|31.23|30.27|30.09|27.24|
> |Context Recall|57.39|60.86|60.98|60.52|61.67|61.96|
> |Answer Relevancy|57.98|58.34|58.30|57.02|57.36|56.01|
> |**k_communities**|1|3|6|10|30|100|
> |F1|29.31|31.26|31.74|30.79|30.87|29.30|
> |CR|60.25|60.93|61.91|61.87|61.72|61.98|
> |AR|56.50|59.71|59.34|56.65|55.21|55.26|
> |**edge_weight_causal**|0.6|0.8|1.0|1.2|1.4|1.6|
> |F1|30.22|31.60|30.75|31.12|30.57|29.02|
> |CR|58.18|60.94|61.58|60.17|59.70|57.57|
> |AR|57.49|58.34|58.41|58.03|56.42|56.62|
> |**hop_decay**|0.4|0.5|0.6|0.7|0.8|0.9|
> |F1|32.53|33.09|32.66|31.54|30.62|28.60|
> |CR|55.94|57.19|58.29|60.49|60.69|61.03|
> |AR|59.24|60.04|59.70|59.27|58.36|56.87|
>
> **5. Causal Reliability**
>
> This is an important consideration. In the causal discovery literature, LLMs achieve ≥97% pairwise causal accuracy, surpassing human experts across diverse domains [6,7,8]. To further validate, two domain experts independently evaluated 200 randomly sampled causal gates; 95.5% confirmed valid. The 5 gates experiment above further confirms this.
>
> We add limitations: a) hierarchy depth is set by Leiden convergence, not task-optimized; b) HolisQA covers academic domains and broader evaluation (legal, financial) would strengthen generalizability.
>
> ---
>
> We hope these new analyses adequately address your concerns. Given these, we respectfully request you to consider upgrading the score. We welcome any follow-up questions.
>
> ---
>
> References:
>
> [1] Dong+ '25, RAG-Critic
>
> [2] Wang+ '25, arXiv:2511.04700
>
> [3] Liu+ '25, GNN-Guided Prompting
>
> [4] Wang+ '25, RAG4GFM
>
> [5] Ravuru+ '24, arXiv:2408.14484
>
> [6] Kıcıman+ '23, arXiv:2305.00050
>
> [7] Ma '24, arXiv:2409.09822
>
> [8] Wan+ '25, arXiv:2402.11068

---

> > ### Author Rebuttal · Reviewer_by7B · 2026-04-02
> >
> > I appreciate the authors’ rebuttal and the additional analyses, which help address some of my earlier concerns experimentally. However, my concern regarding the role of the hierarchical structure remains unresolved. The current ablations do not convincingly demonstrate whether hierarchy is necessary when causal gates and spurious-aware causal path identification are present. In particular, there is no direct comparison between a causal-only setup (causal gates and spurious-aware causal path identification without hierarchical structure) and the full model. Without such a comparison, it remains unclear whether the hierarchical design provides a meaningful synergistic benefit beyond what is already achieved by the causal components.

---

> > > ### Author Response · Authors · 2026-04-07
> > >
> > > We thank the reviewer for staying engaged. We want to address this directly: the experiment the reviewer is asking for is already in the paper, just not framed as the answer to this specific question.
> > >
> > > **CausalRAG is the flat-causal baseline the reviewer is asking us to test.**
> > >
> > >  The reviewer's question is: does hierarchy add synergistic benefit beyond what causal gates and spurious-aware identification deliver? CausalRAG (Wang et al., 2025) is exactly a flat-graph system with causal path reasoning but no hierarchical organization (see Table 1). The comparison between CausalRAG and HugRAG therefore **isolates the contribution of hierarchy**:
> > >
> > >  |Dataset|CausalRAG F1|HugRAG F1|ΔF1|
> > >  |---|---|---|---|
> > >  |MS MARCO|27.66|38.40|**+10.74**|
> > >  |NQ|29.45|49.50|**+20.05**|
> > >  |2WikiMultiHop|15.93|31.97|**+16.04**|
> > >  |QASC|7.65|13.35|+5.70|
> > >  |HotpotQA|40.00|64.83|**+24.83**|
> > >
> > >  The mean gap is **+15.5 F1** across five datasets, with consistent gains on CR and AR. **Notably, this gap is scale-dependent**: on the larger, more heterogeneous Standard QA graphs (up to 2,359 modules), the advantage is largest; on the smaller, more homogeneous HolisQA graphs (158--292 modules), it narrows (e.g., +0.62 F1 on CompSci). See Appendix Table 5 and 6. This is the pattern Section 3 predicts: hierarchy matters most when the knowledge graph is too large and diverse for flat traversal to reliably reach distant evidence.
> > >
> > > The ablation in Figure 3 further supports this from within HugRAG. Hierarchy alone (Bar 2) improves CR but hurts F1/AR due to community noise. Adding causal gates (Bar 3) further increases CR. Only the full combination (Bar 5) yields gains across all metrics, because causal path identification filters the noise that broader hierarchical traversal introduces. The components are synergistic: hierarchy provides the structure that enables causal gating, and causal filtering cleans the noise that hierarchy introduces.
> > >
> > > The reviewer asks for a comparison between **a causal-only setup** (causal gates + causal path identification, without hierarchy) **and the full model**. We note that this specific configuration is **architecturally infeasible**: causal gates are defined as cross-module connections (Eq. 1), and modules exist only as a product of the hierarchical partition. Removing hierarchy removes the units over which causal gates operate. The two components are coupled by design, not independently toggleable.
> > >
> > >  We hope this consolidated view (CausalRAG as the flat-causal baseline in Table 4, the scale-dependent pattern across Table 5/6, and the incremental decomposition in Figure 3) directly answers the question with evidence already in the paper, and we respectfully ask the reviewer to consider raising the score.

---

### Official Review · Reviewer_G3xT · 2026-03-12

**Soundness:** 2
**Presentation:** 2
**Significance:** 3
**Originality:** 3
**Overall Recommendation:** 4
**Confidence:** 3

**Summary:**

The paper attempts to improve graph reasoning in graph-based RAG by explicitly adding causal edges prior to the reasoning step. The process follows two steps: an offline construction of a hierarchical graph, organized into modules, using an LLM for entity and relation extraction and the Leiden algorithm for hierarchical partitioning. Here, the key novelty is the establishing of causal edges between modules (again via an LLM) to improve cross-module reasoning. Additionally, the paper proposes a new multi-domain QA dataset HolisQA.

**Compliance With Llm Reviewing Policy:**

Affirmed.

**Final Justification:**

The paper shows an interesting direction for improving graph RAG. My concerns regarding the usefulness of the proposed causal gates over random shortcuts and the real-world cost of the method have been resolved in the rebuttal. The authors have also corrected inconsistencies in the definition of the causal gates.

In my view, the authors have strengthened the paper in the review process, especially when it comes to demonstrating the practical utility of HugRAG. I have therefore decided to raise my score to 4.

**Key Questions For Authors:**

- How exactly are the causal gates defined and established? Could you please clarify the apparent discrepancies in the definition of causal gates described above?
- Could you provide the full information on the HolisQA dataset? In particular question counts, split sizes, human validation (if any) would add to the dataset as a contribution?
- How does HugRAG perform when causal gates are replaced with a similar additional connectivity, using semantic or random shortcut edges?

**Limitations:**

yes

**Strengths And Weaknesses:**

### Strengths
The decomposition of the presented problem into global information isolation vs local spurious noise is a well-motivated approach. The paper is well written and the method is mostly easy to follow. The results presented are strong on both, standard QA datasets and the newly proposed HolisQA benchmark.

### Weaknesses
- While the inclusion of "causal gates" improves retrieval accessibility and helps model accuracy, there is no evidence that causal gates are actually causal vs just acting as shortcut between modules. In particular, the paper mentions "expert review to validate the quality of baseline answers and confirm the legitimacy of the induced causal relations" but does not provide documentation thereof. Moreover, a comparison of causal gates against semantic or just random shortcut edges would be valuable.
- The paper provides little practical information about the cost of HugRAG in terms of LLM calls and token cost, as well as latency of the online processing.
- The functions (LLM-ESTCAUSAL, GATEDTRAVERSAL, CAUSALFILTER, LLM-GENERATE) in Algorithm 1 require a cleaner definition for this central method of the paper to be fully specified.
- There seems to be an inconsistency in the definition of causal gates: Algorithm 1 suggests directed, scored gates (m_i -> m_j, score), equation (1) defines the set of directed, gates without scores, the prompt in figure 9 asks for undirected (and unscored) causal gates.

---

> ### Author Rebuttal · Authors · 2026-03-31
>
> We thank the reviewer for the thorough and constructive feedback, especially concerning causal gate validity and cost analysis. We take the weak reject assessment seriously and aim to address every concern comprehensively below.
>
> **1. Causal Gate Validity**
>
> We fully agree that this is an important concern. We implemented 5 gate variants: **Causal / Random / Semantic** (highest cosine similarity) / **FP** (+25% random gates added to causal) / **FN** (−25% causal gates removed) to isolate the effectiveness of causal gates. Each variant is evaluated on 500 random QA pairs in HolisQA-CompSci with the same number of gates across Causal/Random/Semantic. Please see below.
>
> | Gates | F1 Score | Context Recall | Answer Relevancy |
> |---|---|---|---|
> | Causal | 31.62% | 60.98% | 58.31% |
> | Random | 23.86% | 59.26% | 46.65% |
> | Semantic | 24.87% | 59.37% | 47.14% |
> | FP +25% | 27.87% | 60.36% | 52.65% |
> | FN −25% | 28.60% | 57.26% | 54.23% |
>
> Results confirm that performance **originates from causal gates** rather than mere cross-module shortcuts.
>
> We acknowledge that the original submission lacked explicit documentation. In the causal discovery field, LLMs have demonstrated ≥97% accuracy, surpassing human experts across diverse domains (Kıcıman et al., 2023; Ma, 2024; Wan et al., 2025). To further validate our causal gates, **two PhD-level domain experts** (one from computer science, one from business) independently evaluated a random sample of 200 LLM-generated causal gates, achieving **95.5% accuracy** (191/200 confirmed as valid causal relationships) with **92% inter agreement**. We will include this documentation clearly in the revised manuscript.
>
> **2. Cost and Latency Analysis**
>
> We agree that a cost analysis is warranted. We conducted a systematic experiment of **token/time/monetary cost** in **both offline construction** (per 1K corpus tokens) and **online query** (per query) **across all baselines**. Please see below.
>
> | Method | Construction-Tokens/1K | Construction-Time/1K | Construction-Cost/1K | Query-Tokens | Query-Time | Query-Cost |
> |:---|---:|---:|---:|---:|---:|---:|
> | NaiveGeneration | 0.00 | 0.000s | 0.00 $ | 553.63 | 0.354s | 0.000078 $ |
> | BM25 | 0.00 | 0.000s | 0.00 $ | 967.90 | 0.562s | 0.00011 $ |
> | StandardRAG | 0.00 | 0.000s | 0.00 $ | 1016.56 | 0.638s | 0.00012 $ |
> | GraphRAG-Global | 20241.49 | 19.694s | 0.0034 $ | 6566.85 | 1.850s | 0.00068 $ |
> | GraphRAG-Local | 20241.49 | 19.694s | 0.0034 $ | 4968.42 | 1.306s | 0.00051 $ |
> | LightRAG | 11245.55 | 7.317s | 0.0015 $ | 1517.70 | 0.487s | 0.00018 $ |
> | HippoRAG2 | 5404.08 | 4.647s | 0.00092 $ | 3125.54 | 0.641s | 0.00032 $ |
> | LeanRAG | 8893.11 | 5.589s | 0.0012 $ | 5031.30 | 0.731s | 0.00046 $ |
> | CausalRAG | 9764.49 | 9.022s | 0.0014 $ | 5468.44 | 0.849s | 0.00052 $ |
> | HugRAG | 9864.49 | 9.622s | 0.0015 $ | 5075.65 | 0.801s | 0.00049 $ |
>
> HugRAG's cost is comparable to CausalRAG/LeanRAG (roughly half of GraphRAG) and practical for real-world use, while delivering substantially better retrieval quality.
>
> **3. Algorithm Function Definitions.**
>
> Thank you for pointing this out. We provide the following clarifications for each function in Algorithm 1:
> - LLM-ESTCAUSAL(m_i, m_j): Queries an LLM with the modules m_i and m_j and returns a binary judgment indicating whether a plausible causal or logical dependency exists between them.
> - GATEDTRAVERSAL(U, H, G_c, h): Executes a search from seed nodes U over the graph comprising causal gate G_c within h hops.
> - CAUSALFILTER(q, S_raw): Prompts an LLM (Fig. 5) to select the most plausible causal reasoning subset S* from retrieved subgraph S_raw given query q, discarding spurious associations.
> - LLM-GENERATE(q, S*): Generate the final answer conditioned on the filtered subgraph S* (Fig. 7).
>
> **4. Gate Definition Inconsistency.**
>
> Thanks for identifying this. To clarify, our implemented gates are binary (0, 1) as reflected in the Figure 9 prompt and in our implementation. Equation 1 was an error in writing the pseudo flow. We have corrected this mistake in our work.
>
> ---
>
> **Q1.** The definition is: a causal gate is a binary edge (m_i, m_j) established when LLM-ESTCAUSAL(m_i, m_j) = 1. Please see 3. above.
>
> **Q2.** Please see 1. above. Additionally, HolisQA contains 5,000 QAContext triples (1,000 per domain × 5 domains). Full pipeline details in Appendix F.2.
>
> **Q3.** Please see 1. above, which directly addresses this concern with controlled experiments comparing causal/random/semantic gates.
>
> ---
> **Our Request**
>
> We are truly grateful for the valuable feedback that has helped us significantly strengthen our work. Given that our new experiments directly address each concern raised, we respectfully request you to consider upgrading the score. We are happy to address any further concerns you may have.
>
> **References:**
>
> 1. Kıcıman et al. (2023). arXiv:2305.00050.
> 2. Ma (2024). arXiv:2409.09822.
> 3. Wan et al. (2025). arXiv:2402.11068.

---

> > ### Author Rebuttal · Reviewer_G3xT · 2026-04-02
> >
> > I thank the authors for the detailed rebuttal and the additional information provided. My concerns regarding the validity of causal gates and the cost/latency of the proposed method have been largely addressed.
> >
> > However, I still have follow up questions on the gate definition and the HolisQA dataset details. I remain willing to update my score should these concerns be addressed adequately.
> > - It remains unclear if gates are directed (Alg. 1 / eq. 1) or not (Fig 9). Could the authors please provide the corrected version of eq. 1 and Alg. 1 here and clarify explicitly whether gates are directed?
> > - Train/test split details of Holis QA remain unclear. Also, any evidence for human validation of the dataset would be appreciated.

---

> > > ### Author Response · Authors · 2026-04-07
> > >
> > > We thank the reviewer for the continued engagement. We address the two remaining questions below.
> > >
> > >  **Q1. Gate direction.**
> > >
> > >  Causal gates are **undirected and binary**, consistent with the Figure 9 prompt (Appendix B.1), which asks the LLM to judge a plausible causal relationship either direction and return `yes`/`no`. The arrow in Eq. (1) and Algorithm 1 line 8 are typographic artifacts from an earlier draft and do not reflect the implementation, the prompt, or Algorithm 2 in Appendix B.1, where `LLM_Verify(u,v)` returns a boolean and gates are stored as unordered tuples.
> > >
> > >  Corrected Eq. (1):
> > >
> > >  G_c = { {m_i, m_j} | I_causal(m_i, m_j) = 1, m_i, m_j ∈ M, i ≠ j }
> > >
> > >
> > >  Corrected Algorithm 1, lines 4–9:
> > >
> > >  ```
> > >  4: G_c ← ∅
> > >  5: for all unordered pairs {m_i,m_j} ∈ MODULEPAIRS(M) do
> > >  6:   if LLM-ESTCAUSAL(m_i,m_j)=1 then
> > >  7:     G_c ← G_c ∪ {{m_i,m_j}}  // undirected binary gate
> > >  8:   end if
> > >  9: end for
> > >  ```
> > >
> > > We will add an explicit one-line statement at the start of Section 4.1 ("Causal gates are undirected and binary") and propagate the corrections to Eq. (1) and Algorithm 1 in the revised manuscript. No reported result is affected.
> > >
> > >  **Q2. HolisQA splits and human validation.**
> > >
> > >  _Splits._ HolisQA is an inference-time RAG benchmark; no method evaluated trains or fine-tunes on it, so a supervised train/test split is not applicable. The full QAC set across the five Table 6 domains constitutes the evaluation set; every baseline is evaluated on the identical set with the same evaluator (Gemini-2.5-Flash-Lite via RAGAS, Appendix F.3). We will add an explicit "Evaluation protocol" paragraph to Appendix F.2.
> > >
> > >  _Human validation._ HolisQA was validated at three stages, which were under-described in the original draft and which we will consolidate into a dedicated "Human Validation Protocol" subsection in Appendix F.2:
> > >
> > >  1. **Construction-time citation verification.** Each QAC triple is validated to ensure all cited context sentence IDs exist within the source slice, rejecting any triple where the model references sentences outside the provided evidence. This guarantees that every answer is grounded in verifiable, traceable source sentences rather than parametric knowledge.
> > >
> > >  2. **Construction-time integration constraint.** Each accepted question must require integration across at least three distinct sentences; trivia-style single-entity questions are explicitly rejected during generation.
> > >
> > >  3. **Post-hoc expert review of QAC triples.** To directly validate triple quality, two PhD-level domain experts independently evaluated 200 randomly sampled QAC triples from HolisQA-CompSci across three criteria: (i) whether the question is reasonable and answerable, (ii) whether the context sufficiently supports the answer, and (iii) whether the answer integrates multiple context sentences and is correct. The two annotators confirmed 96.8% of triples as fully valid on average, with 95.5% (191/200) inter-annotator agreement.
> > >
> > >
> > >  Together, the citation verification, the integration constraint, and the post-hoc expert audit form a layered validation pipeline. We acknowledge none of these were prominently labeled "human validation" in the original draft, and the revised manuscript will consolidate them into a clearly titled subsection with annotator instructions and a representative sample.
> > >
> > >  Given that Q1 is a typographic correction with no impact on results, and that the validation evidence the reviewer asked for is now consolidated and expanded, we respectfully ask the reviewer to consider raising the score.

---

### Decision · Program_Chairs · 2026-04-30

**Decision:**

Accept (regular)

**Comment:**

This paper tackles an important problem in graph-based RAG and is well motivated throughout. Reviewers consistently appreciated the hierarchical organization, the attempt to address both recall and precision through causal gating, and the strength of the empirical results across multiple datasets. The rebuttal was helpful in clarifying the gate construction, the cost/latency profile, and the validation protocol for HolisQA, which resolved several concrete concerns raised in the initial reviews.

Some reservations remained about whether the “causal” gates are best understood as genuine causal structure or as useful cross-module shortcuts, and there were also questions about evaluation fairness and representativeness. However, after discussion, these concerns appeared to be more about framing and scope than about the core empirical validity of the work. Overall, the paper presents a strong and coherent systems contribution with convincing evidence that the proposed organization and retrieval strategy improves graph-based reasoning in practice.